# Manifold-Optimal Guidance: A Unified Riemannian Control View of Diffusion Guidance

**Zexi Jia** [1]  **Pengcheng Luo** [1]  **Zhengyao Fang** [1]  **Jinchao Zhang** [1]  **Jie Zhou** [1]

## Abstract

Classifier-Free Guidance (CFG) serves as the de facto control mechanism for conditional diffusion, yet high guidance scales notoriously induce oversaturation, texture artifacts, and structural collapse. We attribute this failure to a geometric mismatch: standard CFG performs Euclidean extrapolation in ambient space, inadvertently driving sampling trajectories off the high-density data manifold. To resolve this, we present Manifold-Optimal Guidance (MOG), a framework that reformulates guidance as a local optimal control problem. MOG yields a closed-form, geometry-aware Riemannian update that corrects off-manifold drift without requiring retraining. Leveraging this perspective, we further introduce Auto-MOG, a dynamic energy-balancing schedule that adaptively calibrates guidance strength, effectively eliminating the need for manual hyperparameter tuning. Extensive validation demonstrates that MOG yields superior fidelity and alignment compared to baselines, with virtually no added computational overhead.

## 1. Introduction

Diffusion probabilistic models (Sohl-Dickstein et al., 2015; Ho et al., 2020; Song et al., 2021) have become a central paradigm for generative modeling, enabling high-fidelity synthesis of images, videos, and audio from semantic conditions. In practice, *classifier-free guidance* (CFG) (Ho & Salimans, 2022) is the standard mechanism for steering generation: it amplifies conditional information by linearly extrapolating between unconditional and conditional score estimates. Controlled by a single guidance scale $w$, CFG provides a convenient knob that trades diversity for alignment and is widely used in modern systems (Rombach et al.,

[1]WeChat AI, Tencent Inc., China. Correspondence to: Jinchao Zhang <dayerzhang@tencent.com>.

*Proceedings of the 43rd International Conference on Machine Learning*, Seoul, South Korea. PMLR 306, 2026. Copyright 2026 by the author(s).

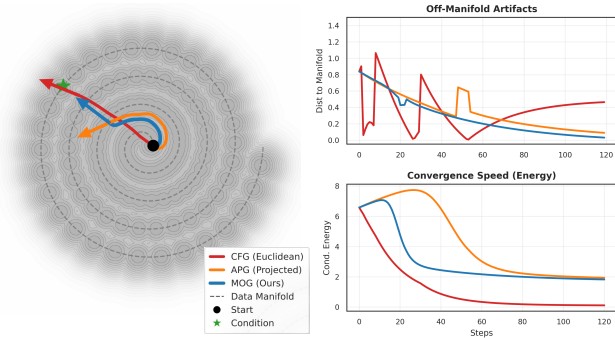

*Figure 1.* Toy spiral manifold illustration of guidance geometry. **Left:** trajectories on a high-density spiral tube. Standard CFG (red) takes a Euclidean shortcut that leaves the manifold. A projected update (orange) stays closer to the manifold but progresses more slowly. MOG (blue) follows a geometry-aware direction that remains near the manifold while moving efficiently toward the condition. **Top right:** distance to the manifold over steps, highlighting off-manifold deviation under CFG. **Bottom right:** conditional energy over steps, showing that MOG reduces energy rapidly without leaving the high-density region.

2022; Ramesh et al., 2022; Peebles & Xie, 2023).

Despite its simplicity, CFG implicitly relies on a strong geometric approximation. It treats the latent space as Euclidean and isotropic, and assumes the conditional increment is a universally valid direction in the ambient space. This approximation works well at moderate guidance, but becomes brittle at high scales. Recent analyses indicate that aggressive extrapolation can push trajectories into low-density regions where the model is forced to extrapolate, leading to oversaturation, unnatural textures, and even structural collapse (Lin et al., 2024). We refer to this failure mode as *off-manifold drift*: stronger guidance can improve alignment by moving faster in the ambient space, yet it may do so by departing from the high-density set defined by the data distribution.

Existing remedies largely address symptoms rather than the underlying geometry. Dynamic thresholding clips extreme values to stabilize sampling (Saharia et al., 2022). Other approaches reshape the guidance signal through hand-designed decompositions or constraints, including projection-based filtering (Sadat et al., 2025) and manifold-motivated cor-

rections (Chung et al., 2025). While effective in specific regimes, these methods are typically motivated as post hoc adjustments. They do not provide a single objective that (i) determines the guidance direction in a manner consistent with data geometry and (ii) prescribes an appropriate update strength during sampling.

In this work, we derive guidance from a principled optimization view. Conditioning induces an energy landscape, and guided sampling can be interpreted as decreasing conditional energy while remaining close to the data manifold. When the geometry is nontrivial, the Euclidean gradient is generally *not* the steepest-descent direction relative to the manifold. Instead, the appropriate direction is the *Riemannian* steepest descent induced by a metric that penalizes motions that move off-manifold. This perspective leads to *Manifold-Optimal Guidance* (MOG): we formulate guidance as a local optimal control problem under a geometry-aware metric and obtain a closed-form update that naturally suppresses off-manifold components while preserving efficient descent.

Figure 1 illustrates the key geometry. Standard CFG takes a Euclidean shortcut that quickly exits the high-density spiral tube, causing large off-manifold deviation. A purely projected update adheres more closely to the manifold but discards useful descent components and slows progress. MOG instead preconditions the conditional increment with a Riemannian metric, attenuating off-manifold directions while retaining effective progress along the manifold. This toy behavior mirrors practical failures under large CFG scales and motivates a geometry-consistent design.

We instantiate MOG with two practical variants used throughout the paper. *MOG-Score* specifies a score-based anisotropic metric with efficient matrix-free inverse application. *Auto-MOG* derives an energy-balanced adaptive scaling rule that sets guidance strength over time, reducing manual tuning.

Our contributions are:

- We formulate guidance as a variational optimization problem under a Riemannian metric and derive MOG as a natural-gradient update that is optimal with respect to a geodesic cost.

- We propose two practical instantiations, MOG-Score and Auto-MOG, that are training-free, efficient, and compatible with standard diffusion samplers.

- We demonstrate consistent improvements across diffusion architectures and benchmarks via quantitative evaluation and a user preference study, achieving better fidelity and alignment with substantially fewer oversaturation and texture artifacts under strong guidance.

**Conflict of Interest Disclosure.** The authors declare no financial conflicts of interest related to this work. The authors are affiliated with Tencent, but this employment does not by itself constitute a conflict of interest; the evaluated models are public research or open models and are not developed by an employer of the authors.

## 2. Related Work

Classifier-Free Guidance (CFG) interpolates between unconditional and conditional score estimates via a single guidance scale (Ho & Salimans, 2022), powering state-of-the-art text-to-image (Rombach et al., 2022; Ramesh et al., 2022; Saharia et al., 2022) and video generation (Ho et al., 2022). However, CFG assumes isotropic Euclidean geometry, and at high guidance scales, trajectories deviate from the data manifold, causing oversaturation and artifacts (Lin et al., 2024; Chung et al., 2025).

**Stabilizing High-Scale Guidance.** Numerous training-free strategies have been proposed to mitigate CFG instabilities. Dynamic Thresholding (Saharia et al., 2022) clips extreme pixel values to suppress saturation. Frequency-domain methods (Song & Lai, 2025) attribute artifacts to low-frequency accumulation and apply selective spectral filtering. Projection-based approaches such as APG (Sadat et al., 2025) decompose the guidance signal and remove components aligned with unconditional predictions. DSG (Yang et al., 2024) introduces spherical Gaussian constraints to bound guidance steps within intermediate manifolds. While effective in specific regimes, these methods rely on hand-crafted heuristics tailored to particular failure modes. They do not provide a unified objective that jointly determines optimal direction and adaptive strength.

**Manifold-Aware Diffusion Guidance.** A growing body of work attributes CFG failures to off-manifold drift. CFG++ (Chung et al., 2025) reformulates guidance updates to respect manifold constraints implied by the forward process, reducing extrapolation errors. Schedule-based interventions, including rectified guidance (Xia et al., 2024) and truncated CFG (Fu & Li, 2025), modify when and how strongly guidance is applied to prevent late-stage instabilities. S-CFG (Shen et al., 2024) addresses spatial inconsistency by introducing location-adaptive scales. ECM Guidance (Wang et al., 2024) explicitly models the clean data manifold as a proxy for controllable generation. These approaches share the motivation of keeping samples near high-density regions but typically address direction correction or strength scheduling in isolation, rather than unifying both within a single principled framework.

**Geometric and Control-Theoretic Perspectives.** Riemannian score-based models (De Bortoli et al., 2022) extend diffusion to curved manifolds, while variational for-

mulations (Pandey et al., 2025) and Lyapunov-based controllers (Mukherjee et al., 2024) cast guidance as optimal control. Schrödinger bridge methods (Liu et al., 2023) connect diffusion to entropy-regularized transport. These frameworks provide elegant foundations but often require specialized training or modifications incompatible with pretrained models.

Our Manifold-Optimal Guidance (MOG) formulates guidance as local optimal transport with anisotropic Riemannian cost, yielding a closed-form solution requiring no retraining. This further yields Auto-MOG, an energy-balancing rule that adaptively calibrates guidance strength, eliminating manual tuning while improving high-scale robustness.

# 3. Manifold-Optimal Guidance

We first summarize the logic of our derivation before introducing the formal notation. Standard CFG applies a Euclidean extrapolation in the ambient latent space. We instead view each guidance step as a local control problem: the update should decrease the conditional energy while paying the transport cost induced by the local data geometry. Introducing a positive-definite Riemannian metric leads to a closed-form natural-gradient update. This view recovers CFG as the Euclidean special case, while MOG follows the steepest descent direction under the geometry encoded by the metric. Appendix B provides the full derivation and the constrained optimality proof.

## 3.1. Preliminaries and Problem Setup

We first introduce notation for diffusion sampling and conditional guidance. Consider diffusion models described by the probability flow ODE. Let $x_t \in \mathbb{R}^d$ denote the latent state at time $t \in [0, T]$, evolving as

$$\frac{\mathrm{d}x_t}{\mathrm{d}t} = f(x_t, t) - g^2(t)\, s_\theta(x_t, t), \qquad (1)$$

where $f(\cdot, t)$ is the drift coefficient, $g(t)$ is the diffusion coefficient, and $s_\theta(x_t, t) \approx \nabla_x \log p_t(x_t)$ is the learned score that approximates the gradient of the log density.

For conditional generation with condition $c$, classifier free guidance provides two score estimates. We denote the unconditional score by $s_0(x_t, t) \triangleq s_\theta(x_t, t, \varnothing)$ and the conditional score by $s_c(x_t, t) \triangleq s_\theta(x_t, t, c)$. Their difference defines the conditional increment

$$\Delta s(x_t, t) \triangleq s_c(x_t, t) - s_0(x_t, t). \qquad (2)$$

Standard classifier free guidance (CFG) (Ho & Salimans, 2022) constructs the guided score by linear extrapolation, $s_{\mathrm{CFG}} = s_0 + w \cdot \Delta s$, where $w \geq 1$ is the guidance scale. This Euclidean combination treats all directions in latent

space as equally valid. In practice, it ignores the non-Euclidean geometry induced by the data manifold $\mathcal{M}$, and large $w$ often causes the sampling trajectory to drift away from $\mathcal{M}$, producing oversaturation and structural artifacts (Fig. 1). This motivates a geometric formulation in which guidance is derived as an optimal update under the intrinsic structure of the data distribution.

## 3.2. Riemannian Optimal Control for Guidance

We derive MOG by casting guidance as a local optimal control problem on the data manifold. Given condition $c$, the goal is to modify the score in a way that improves alignment while avoiding off-manifold drift.

We associate $c$ with a potential energy $\mathcal{E}(x_t, c) \triangleq -\log p_t(c \mid x_t)$, which measures how well $x_t$ aligns with $c$. By Bayes' rule, up to an additive constant independent of $x_t$, the conditional score decomposes as

$$\nabla_x \log p_t(x_t \mid c) = \nabla_x \log p_t(x_t) + \nabla_x \log p_t(c \mid x_t), \qquad (3)$$

which implies that the conditional increment approximates the negative energy gradient,

$$\Delta s(x_t, t) \approx -\nabla_x \mathcal{E}(x_t, c). \qquad (4)$$

Guidance can therefore be viewed as energy minimization, descending the conditional landscape while staying close to the data manifold.

We seek an update $u_t \in \mathbb{R}^d$ that defines a guided score $s_{\mathrm{guid}} = s_0 + u_t$. The update should reduce $\mathcal{E}(x_t, c)$ while respecting local geometry. We encode geometry through a Riemannian metric $\mathbf{M}_t(x_t) \succ 0$ and define the local objective

$$\mathcal{J}_t(u_t) = \underbrace{\frac{1}{2} u_t^\top \mathbf{M}_t u_t}_{\text{Transport Cost}} + \underbrace{\beta(t) \langle \nabla_x \mathcal{E},\, u_t \rangle}_{\text{Alignment Objective}}, \qquad (5)$$

where $\beta(t) \geq 0$ controls the guidance strength. The quadratic term penalizes movement that is costly under the metric, while the linear term encourages decreasing the conditional energy. Equivalently, this Lagrangian form corresponds to maximizing instantaneous energy decrease under a fixed $\mathbf{M}_t$-geodesic budget; the constrained proof is given in Appendix B.2.

Since $\mathbf{M}_t \succ 0$, the objective is strictly convex and has a unique minimizer. Setting $\nabla_{u_t} \mathcal{J}_t = \mathbf{M}_t u_t + \beta(t) \nabla_x \mathcal{E} = 0$ yields

$$u_t^\star = -\beta(t)\, \mathbf{M}_t^{-1} \nabla_x \mathcal{E}(x_t, c). \qquad (6)$$

Substituting Eq. (4) gives the Manifold-Optimal Guidance update

$$\boxed{s_{\mathrm{MOG}} = s_0 + \beta(t)\, \mathbf{M}_t^{-1} \Delta s.} \qquad (7)$$

Eq. (7) can be interpreted as Riemannian natural gradient descent. In Euclidean space, steepest descent follows $-\nabla_x \mathcal{E}$. Under a nontrivial geometry, the steepest descent direction becomes the Riemannian gradient $\mathrm{grad}_\mathcal{M} \mathcal{E} = \mathbf{M}_t^{-1} \nabla_x \mathcal{E}$. The inverse metric $\mathbf{M}_t^{-1}$ thus preconditions the update to respect manifold geometry, amplifying components consistent with the manifold and suppressing directions that induce drift.

### 3.3. Unified View and Optimality Analysis

We next present a unified variational view of guidance and characterize the optimality of MOG. Consider the generalized optimization problem

$$u^\star(\mathbf{Q}) = \underset{u \in \mathbb{R}^d}{\arg\min} \ \frac{1}{2} u^\top \mathbf{Q} u + \beta \langle \nabla_x \mathcal{E}, u \rangle, \qquad (8)$$

where $\mathbf{Q} \succ 0$ defines the quadratic cost. Different choices of $\mathbf{Q}$ correspond to different geometric assumptions about the latent space. CFG is recovered by setting $\mathbf{Q} = \mathbf{I}$, which treats all directions equally and yields $u^\star \propto \Delta s$. Other guidance rules can be interpreted as adding quadratic penalties on specific directions. For example, CFG++ penalizes radial motion through $\mathbf{Q} = \mathbf{I} + \lambda \hat{x} \hat{x}^\top$ with $\hat{x} = x_t / \|x_t\|$, and APG penalizes updates aligned with the unconditional score through $\mathbf{Q} = \mathbf{I} + \lambda s_0 s_0^\top$ as $\lambda \to \infty$, which projects updates onto the subspace orthogonal to $s_0$.

The principled choice of $\mathbf{Q}$ is given by the intrinsic geometry of the data distribution. Latent states concentrate near a lower dimensional manifold $\mathcal{M}$, and the corresponding natural cost is induced by the Riemannian metric $\mathbf{M}_t$. Setting $\mathbf{Q} = \mathbf{M}_t$ yields MOG as the unique minimizer of Eq. (8).

We further show that MOG achieves the maximum energy reduction per unit geodesic displacement. Consider

$$v^\star = \underset{\|v\|_{\mathbf{M}_t} \leq 1}{\arg\min} \langle \nabla_x \mathcal{E}, v \rangle, \qquad (9)$$

where $\|v\|_{\mathbf{M}_t} = \sqrt{v^\top \mathbf{M}_t v}$ measures geodesic length. By the generalized Cauchy Schwarz inequality $|\langle x, y \rangle| \leq \|x\|_{\mathbf{M}_t^{-1}} \|y\|_{\mathbf{M}_t}$,

$$\langle \nabla_x \mathcal{E}, v \rangle \geq -\|\nabla_x \mathcal{E}\|_{\mathbf{M}_t^{-1}} \cdot \|v\|_{\mathbf{M}_t}. \qquad (10)$$

Equality holds if and only if $v \propto -\mathbf{M}_t^{-1} \nabla_x \mathcal{E}$. Hence, MOG follows the unique steepest descent direction under the manifold metric. Defining the guidance efficiency $\eta(u) \triangleq -\langle \nabla_x \mathcal{E}, u \rangle / \|u\|_{\mathbf{M}_t}$, MOG achieves the maximum $\eta_{\mathrm{MOG}} = \|\nabla_x \mathcal{E}\|_{\mathbf{M}_t^{-1}}$. In contrast, CFG (which uses $u \propto -\nabla_x \mathcal{E}$) generally attains lower efficiency under an anisotropic metric $\mathbf{M}_t$, which explains why it requires larger, more artifact-prone updates to reach comparable alignment.

## 4. Practical Instantiation

### 4.1. MOG-Score: Score-Based Anisotropic Metric

The framework in Section 3 requires a metric tensor $\mathbf{M}_t$, but explicitly storing a $d \times d$ matrix is infeasible in high dimensions. Useful metrics often admit low rank or diagonal structure, enabling $O(d)$ matrix vector products without explicit construction. We instantiate $\mathbf{M}_t$ using quantities already available during sampling.

The unconditional score $s_0 = \nabla_x \log p_t(x_t)$ points toward higher density regions and serves as a local proxy for the normal direction to the noisy data manifold. This motivates an anisotropic metric that distinguishes normal motion, which tends to leave the high-density region, from tangential motion, which moves along the manifold.

We define the unit normal $n_t = s_0 / \|s_0\|$ and construct a full-rank positive-definite metric with eigenvalue $\lambda_\perp$ for the estimated normal direction and $\lambda_\parallel$ for tangent directions. Choosing $\lambda_\perp \gg \lambda_\parallel$ penalizes off-manifold motion while allowing tangential movement. The metric is a rank-one anisotropic update to a scaled identity, $\mathbf{M}_t = \lambda_\parallel \mathbf{I} + (\lambda_\perp - \lambda_\parallel) n_t n_t^\top$, and admits a closed-form inverse via Sherman–Morrison:

$$\mathbf{M}_t^{-1} v = \frac{1}{\lambda_\parallel} v - \frac{\lambda_\perp - \lambda_\parallel}{\lambda_\parallel \lambda_\perp} (n_t^\top v) \, n_t. \qquad (11)$$

This computation uses one inner product and a scaled addition, yielding $O(d)$ complexity. We set $\lambda_\parallel = 1.0$ and the anisotropy ratio $\rho = \lambda_\perp / \lambda_\parallel = 10$ throughout.

### 4.2. Auto-MOG: Energy-Balanced Adaptive Scaling

MOG uses a scalar weight $\beta(t)$ in $s_{\mathrm{MOG}} = s_0 + \beta(t) \mathbf{M}_t^{-1} \Delta s$, where $\Delta s = s_\theta(x_t, t, c) - s_0$ is the standard conditional residual in CFG. A fixed $\beta(t)$ can be brittle across noise levels and prompts. Auto-MOG derives $\beta(t)$ from an energy balance principle under $\mathbf{M}_t$. We use the following naming convention: MOG-Score corrects the guidance direction while retaining a user-specified scale, whereas Auto-MOG uses the same geometry-aware direction and additionally adapts the scale over time.

Auto-MOG only requires matrix-free application of $\mathbf{M}_t^{-1}$ and quadratic forms under $\mathbf{M}_t$. In our implementation, Auto-MOG utilizes a combined metric estimator that integrates both the score-based anisotropic metric and a feature-diagonal approximation. For the latter, we extract feature maps $h_t \in \mathbb{R}^{C \times H \times W}$ from the final layer of the denoiser and compute per-channel spatial variance $\sigma_c^2(t) = \mathrm{Var}_{i,j}[h_{t,c,i,j}] + \epsilon$. To align with the latent geometry, this $C$-dimensional variance vector is spatially broadcast to match the latent dimension $d$, yielding a diagonal metric applied via efficient element-wise rescaling. In ablation studies, we also evaluate these estimators individually.

**Algorithm 1** MOG-Score and Auto-MOG
---

1: **Input:** Initial noise $x_T$, condition $c$, steps $N$, balance factor $\gamma$
2: **Output:** Generated sample $x_0$
3: **for** $n = N, N - 1, \ldots, 1$ **do**
4:     $s_0 \leftarrow s_\theta(x_t, t, \varnothing)$
5:     $\Delta s \leftarrow s_\theta(x_t, t, c) - s_0$
6:     // Metric inverse application (Eq. 11)
7:     $u \leftarrow \mathbf{M}_t^{-1}[\Delta s]$
8:     // Auto-MOG scaling (Eq. 14)
9:     $\beta_t \leftarrow \gamma \cdot \|s_0\|_{\mathbf{M}_t} / (\|u\|_{\mathbf{M}_t} + \varepsilon)$
10:     $x_{t-1} \leftarrow \text{SAMPLER}(x_t, s_0 + \beta_t \cdot u, t)$
11: **end for**
12: **return** $x_0$

---

Effective guidance should neither overwhelm the prior score nor be negligible. Let $v_{\text{nat}} = \mathbf{M}_t^{-1} \Delta s$ denote the natural gradient direction. We require the guidance energy to remain proportional to the prior energy:

$$\left\| \beta(t) \, v_{\text{nat}} \right\|_{\mathbf{M}_t} = \gamma \, \|s_0\|_{\mathbf{M}_t}, \tag{12}$$

where $\|v\|_{\mathbf{M}_t} = (v^\top \mathbf{M}_t v)^{1/2}$ and $\gamma > 0$ is a normalized balance factor controlling the target guidance-to-prior energy ratio, rather than a numerical copy of the CFG scale. Squaring both sides and using $v_{\text{nat}}^\top \mathbf{M}_t v_{\text{nat}} = \Delta s^\top \mathbf{M}_t^{-1} \Delta s$ yields

$$\beta_{\text{auto}}(t) = \gamma \sqrt{\frac{s_0^\top \mathbf{M}_t s_0}{\Delta s^\top \mathbf{M}_t^{-1} \Delta s}}. \tag{13}$$

In practice, defining prior energy $E_{\text{prior}}(t) = \|s_0\|_{\mathbf{M}_t}$ and guidance energy $E_{\text{guid}}(t) = \|v_{\text{nat}}\|_{\mathbf{M}_t}$, this simplifies to

$$\beta_{\text{auto}}(t) = \gamma \cdot \frac{E_{\text{prior}}(t)}{E_{\text{guid}}(t) + \varepsilon}, \tag{14}$$

where $\varepsilon > 0$ ensures stability. For the MOG-Score metric, these energies reduce to projections involving $n_t$, incurring negligible overhead.

This schedule naturally decays during sampling as $\|s_0\|$ decreases with improving signal to noise ratio. It also adapts to prompt difficulty: weak conditional signals yield larger $\beta$ to compensate, while strong signals receive smaller $\beta$ to prevent oversaturation.

### 4.3. Algorithm and Complexity

Algorithm 1 summarizes the MOG framework. At each denoising step, we first compute unconditional and conditional scores as in standard CFG. MOG then applies a matrix-free metric operator to obtain the natural guidance direction via inverse application (Line 7), followed by adaptive scaling through the energy ratio (Line 9). The resulting guided score directly replaces the CFG combination in any standard sampler.

The computational overhead is negligible. MOG-Score requires only a small number of inner products and vector additions per step, yielding $O(d)$ complexity. Auto-MOG additionally evaluates two quadratic forms, maintaining the same $O(d)$ cost. Since network inference dominates overall runtime, the extra computation introduced by MOG is marginal in practice.

## 5. Experiments

### 5.1. Experimental Setup

**Models and Datasets.** We conduct experiments on six representative diffusion models spanning diverse architectures. For class-conditional generation, we employ DiT-XL/2 (Peebles & Xie, 2023) and EDM2-XXL (Karras et al., 2024) on ImageNet (256×256). For text-to-image synthesis, we evaluate Lumina (Liu et al., 2024), SD-2.1 (Rombach et al., 2022), SD-XL (Podell et al., 2024), SD-3.5 (Esser et al., 2024), and FLUX.1 (Black Forest Labs et al., 2025) using prompts from the MS-COCO 2017 (Lin et al., 2014) validation set. This selection encompasses UNet architectures, Transformer-based models (DiT, MMDiT), and Rectified Flow formulations, providing comprehensive coverage of modern generative systems.

**Baselines.** We compare MOG against four representative guidance methods. Standard CFG (Ho & Salimans, 2022) serves as the primary baseline. CFG++ (Chung et al., 2025) applies manifold-motivated radial projection to reduce drift. APG (Sadat et al., 2025) projects guidance orthogonally to the unconditional prediction. LF-CFG (Song & Lai, 2025) filters low-frequency components to attenuate accumulated artifacts. For all baselines, we use official implementations with default hyperparameters.

**Evaluation Metrics.** We adopt a comprehensive evaluation protocol measuring four aspects. For distribution fidelity, we report Fréchet Inception Distance (FID) (Heusel et al., 2017), Precision, and Recall (Kynkäänniemi et al., 2019). For artifact quantification, we measure Saturation (mean HSV saturation) and Contrast (pixel-wise standard deviation) following (Sadat et al., 2025). For text-image alignment, we compute CLIP Score (ViT-g/14) (Radford et al., 2021) and Human Preference Score v2 (HPSv2) (Wu et al., 2023). We additionally report Aesthetic Score (AES) (Schuhmann et al., 2022) in ablation studies.

**Implementation Details.** We use MOG-Score with anisotropy parameter $\rho=10$ for fixed-$\beta$ variants. For Auto-MOG, we employ the combined MOG-Score and MOG-Feat metric by default, setting $\gamma=1$ across all main experiments. Images are generated at native resolution and resized

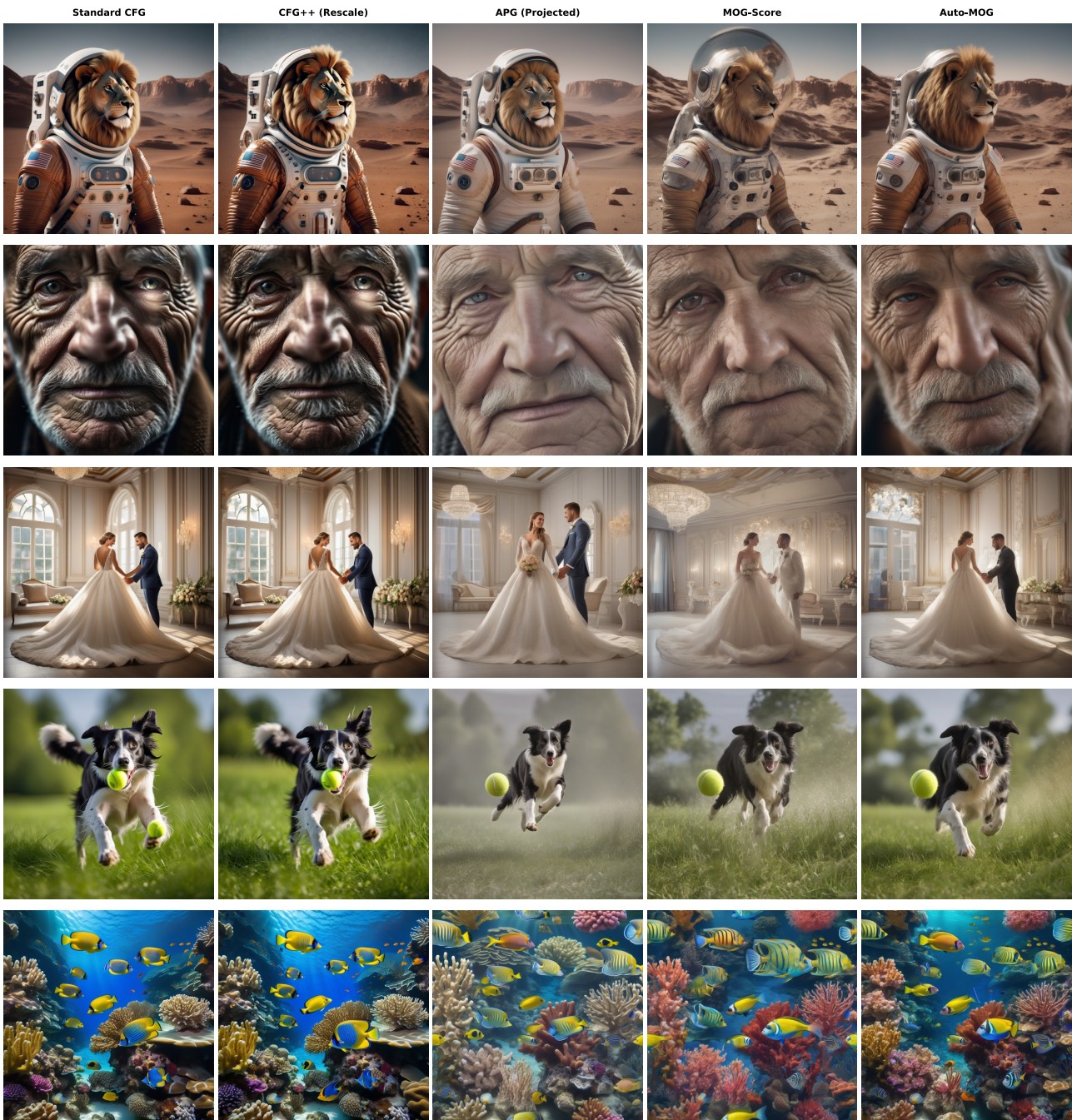

| Standard CFG | CFG++ (Rescale) | APG (Projected) | MOG-Score | Auto-MOG |

*Figure 2.* Qualitative comparison across guidance methods. All images are generated with Stable Diffusion XL at guidance scale 15. Standard CFG manifests oversaturation and harsh textures. CFG++ yields washed-out results with low contrast. APG introduces a hazy appearance. In contrast, Auto-MOG achieves the most realistic results.

to $512{\times}512$ for metric computation. Latency is measured on a single NVIDIA A100 GPU with batch size 1.

### 5.2. Qualitative Evaluation

Visual comparisons at high guidance scales are presented in Figure 2. Standard CFG produces severe artifacts that deviate from the natural image manifold, manifesting as oversaturation and structural distortion. Although existing baselines provide partial relief, CFG++ reduces saturation at the cost of dynamic range, and APG introduces a distinct haze. In contrast, Auto-MOG consistently maintains structural integrity and visual realism.

*Table 1.* **Quantitative comparison across diverse benchmarks.** DiT-XL/2 and EDM2-XXL are evaluated on **ImageNet 256×256**; all text-to-image models are evaluated on **COCO 512×512**.

| Model | Guidance | FID ↓ | Pre. ↑ | Rec. ↑ | Satu. ↓ | Contrast ↓ | HPSv2 ↑ | CLIP ↑ |
|---|---|---|---|---|---|---|---|---|
| DiT-XL/2 (ImageNet-256, $w = 4$) | CFG | 19.14 | **0.92** | 0.35 | 0.37 | 0.25 | 28.10 | 31.00 |
| | CFG++ | 12.45 | 0.90 | 0.45 | 0.34 | 0.23 | 28.05 | 31.02 |
| | APG | 9.34 | 0.89 | 0.56 | 0.30 | 0.20 | 28.00 | 30.98 |
| | LF-CFG | 10.12 | 0.91 | 0.52 | 0.32 | 0.21 | 28.08 | 31.01 |
| | MOG-Score | 9.05 | **0.92** | 0.57 | **0.29** | **0.19** | 28.20 | 31.10 |
| | Auto-MOG | **8.78** | 0.91 | **0.58** | 0.31 | 0.20 | **28.25** | **31.12** |
| EDM2-XXL (ImageNet-256, $w = 2$) | CFG | 8.65 | 0.84 | 0.57 | 0.37 | 0.23 | 28.50 | 31.10 |
| | CFG++ | 6.12 | 0.83 | 0.62 | 0.34 | 0.22 | 28.55 | 31.12 |
| | APG | 4.94 | 0.83 | 0.67 | **0.30** | 0.21 | 28.45 | 31.08 |
| | LF-CFG | 5.50 | 0.84 | 0.64 | 0.33 | **0.20** | 28.52 | 31.11 |
| | MOG-Score | 4.55 | **0.85** | 0.68 | **0.30** | 0.23 | 28.60 | 31.15 |
| | Auto-MOG | **4.30** | 0.84 | **0.69** | 0.32 | 0.21 | **28.65** | **31.18** |
| SD-2.1 (COCO-512, $w = 10$) | CFG | 27.53 | 0.65 | 0.41 | 0.36 | 0.27 | 25.51 | 30.38 |
| | CFG++ | 24.10 | 0.66 | 0.45 | 0.31 | 0.25 | 25.70 | 30.55 |
| | APG | 22.21 | 0.67 | 0.49 | 0.27 | 0.22 | **25.96** | **31.12** |
| | LF-CFG | 23.05 | **0.68** | 0.46 | 0.29 | **0.20** | 25.62 | 30.50 |
| | MOG-Score | 21.70 | 0.66 | 0.51 | 0.25 | 0.21 | 25.88 | 30.78 |
| | Auto-MOG | **21.55** | 0.67 | **0.52** | **0.24** | **0.20** | 25.92 | 30.91 |
| SD-XL (COCO-512, $w = 15$) | CFG | 22.29 | 0.62 | 0.49 | 0.28 | 0.24 | 28.42 | 33.62 |
| | CFG++ | 22.80 | 0.63 | 0.51 | 0.25 | 0.22 | 28.55 | 33.70 |
| | APG | 22.35 | 0.64 | 0.50 | 0.18 | 0.17 | 28.60 | 33.78 |
| | LF-CFG | 22.60 | 0.63 | 0.50 | 0.22 | 0.20 | 28.53 | 33.72 |
| | MOG-Score | 21.75 | **0.65** | 0.53 | 0.19 | 0.17 | 28.78 | 34.05 |
| | Auto-MOG | **21.60** | 0.64 | **0.54** | **0.17** | **0.16** | **29.00** | **34.20** |
| SD-3.5 (COCO-512, $w = 4.5$) | CFG | 22.13 | 0.81 | 0.56 | 0.41 | 0.31 | 29.10 | 34.60 |
| | CFG++ | 20.50 | 0.82 | 0.59 | 0.38 | 0.29 | 29.35 | 34.75 |
| | APG | 18.90 | 0.82 | 0.61 | **0.30** | **0.23** | 29.55 | 34.90 |
| | LF-CFG | 19.45 | 0.81 | 0.58 | 0.36 | 0.28 | 29.40 | 34.82 |
| | MOG-Score | 18.35 | 0.83 | **0.64** | 0.32 | 0.25 | 29.75 | 35.05 |
| | Auto-MOG | **18.10** | **0.84** | 0.63 | 0.31 | 0.24 | **30.05** | **35.20** |
| FLUX.1 (COCO-512, $w = 4.0$) | CFG | 20.80 | 0.85 | 0.60 | 0.35 | 0.28 | 30.15 | 35.80 |
| | CFG++ | 19.42 | 0.86 | 0.62 | 0.33 | 0.26 | 30.22 | 35.85 |
| | APG | 18.15 | 0.86 | 0.65 | **0.28** | **0.22** | 30.40 | 35.92 |
| | LF-CFG | 18.90 | 0.85 | 0.63 | 0.31 | 0.24 | 30.30 | 35.88 |
| | MOG-Score | 18.23 | **0.88** | **0.67** | 0.29 | 0.23 | 30.65 | 36.10 |
| | Auto-MOG | **17.84** | 0.87 | 0.66 | 0.30 | 0.24 | **30.88** | **36.37** |

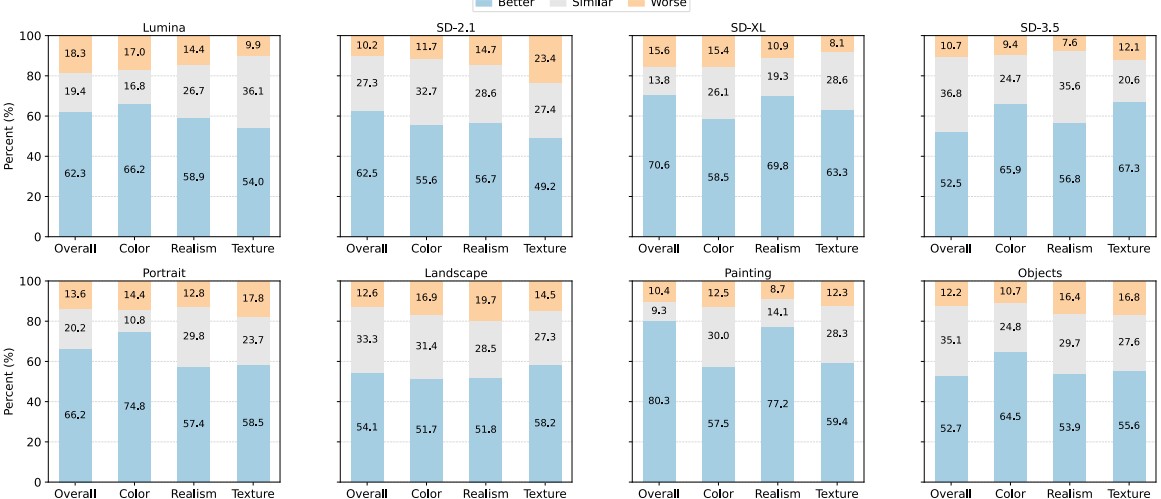

*Figure 3.* User preference study comparing Auto-MOG against standard CFG. Top row: model-based evaluation. Bottom row: domain-based evaluation. Auto-MOG achieves consistent preference advantages, particularly in Color and Realism.

Specific examples illustrate these advantages. In portraiture, Auto-MOG recovers natural skin tones while avoiding the metallic specular highlights typical of CFG. For intricate textures such as the lion astronaut, it preserves fine fur details where CFG collapses into smooth, plastic-like surfaces. In high-contrast scenarios such as the wedding scene, our method prevents highlight clipping. Across diverse subjects ranging from animals to coral reefs, Auto-MOG achieves vibrant yet natural saturation without the detail loss observed in competing methods.

## 5.3. Quantitative Evaluation

Table 1 summarizes quantitative performance across all benchmarks. Auto-MOG consistently establishes state-of-the-art fidelity, achieving an FID of 8.78 on DiT-XL/2 (vs. 12.45 for CFG++) and 4.30 on EDM2-XXL (vs. 4.94 for APG). These improvements extend to the latest architectures: on FLUX.1, Auto-MOG attains an FID of 17.84 compared to 18.15 for APG, while simultaneously achieving superior alignment scores (HPSv2: 30.88, CLIP: 36.37). Such consistent gains across diverse model families suggest that manifold-aware guidance effectively constrains sampling trajectories to high-density regions of the data distribution, independent of the underlying architecture.

Beyond fidelity, MOG demonstrates superior artifact suppression. On SD-XL ($w=15$), standard CFG yields excessive Saturation (0.28) and Contrast (0.24), while Auto-MOG significantly reduces these to 0.17 and 0.16, respectively, aligning closely with natural image statistics. Crucially, this correction does not compromise semantic alignment: Auto-MOG attains the lowest FID while simultaneously achieving the highest HPSv2 (29.00) and CLIP (34.20) scores on SD-XL. This ability to optimize the *alignment-fidelity Pareto frontier* holds consistently across UNet, DiT, MMDiT, and Rectified Flow models, confirming MOG as a unified solution for high-scale guidance.

## 5.4. User Study

To corroborate these objective metrics with human perceptual judgments, we conducted a rigorous blind paired-comparison study ($N = 37$). Participants evaluated 60–80 image pairs per condition across four architectures (Lumina, SD-2.1, SD-XL, SD-3.5), rating outputs on Overall Quality, Color Fidelity, Realism, and Texture.

As shown in Figure 3, Auto-MOG demonstrates clear preference advantages over standard CFG, achieving win rates between 52.5% and 70.6% with rejection rates consistently below 20%. Domain-specific analysis reveals particularly strong performance in *Portraits* (66.2%; 74.8% for Color) and *Paintings* (80.3%), effectively mitigating the specific artifacts that plague baselines in these categories, such as skin tone shift and texture frying. Notable gains are also

observed in *Landscapes* (54.1%) and *Objects* (52.7%). The strong preference in Color and Realism metrics confirms that geometric corrections translate directly to improved perceptual quality. See supplementary for details.

## 5.5. Ablation Studies

We conduct ablation studies on SD-XL (3K images, COCO 2017) to validate design choices.

**Guidance Scale and Steps.** Figure 4 (bottom) shows that MOG-Score exhibits greater stability than CFG at high scales, maintaining higher CLIP and HPSv2 scores. Figure 4 (top) demonstrates that Auto-MOG outperforms fixed-scale methods across all sampling steps (25-50).

**Balance Factor.** Figure 5 shows that performance peaks near $\gamma=1$, with mild degradation elsewhere, indicating that Auto-MOG is robust to hyperparameter tuning.

**Metric Instantiation.** Table 2 compares metric estimators. MOG-Score offers improvements with negligible latency ($1.01\times$). While combining feature-based metrics (MOG-Score+Feat) yields marginal gains, Auto-MOG provides the optimal trade-off between fidelity, alignment, and efficiency.

*Table 2.* Ablation of Riemannian metric estimators on SD-XL.

| Method | FID↓ | HPSv2↑ | CLIP↑ | AES↑ | Latency |
|---|---|---|---|---|---|
| CFG ($M=I$) | 26.29 | 28.42 | 33.62 | 6.65 | $1.00\times$ |
| MOG-Score | 25.75 | 28.88 | 34.35 | 6.70 | $1.01\times$ |
| MOG-Feat | 25.62 | 29.16 | 34.20 | 6.72 | $1.06\times$ |
| MOG-Score+Feat | 25.45 | 29.24 | 34.43 | 6.73 | $1.08\times$ |
| **Auto-MOG** | **25.10** | **29.36** | **34.67** | **6.75** | $1.08\times$ |

*Table 3.* Sensitivity to anisotropy ratio $\rho$ (MOG-Score).

| $\rho$ | FID ↓ | CLIP ↑ | HPSv2 ↑ | Satu. ↓ |
|---|---|---|---|---|
| 1 | 26.20 | 33.70 | 28.45 | 0.27 |
| 5 | 25.85 | 34.30 | 28.82 | 0.20 |
| **10 (Def.)** | **25.75** | **34.35** | **28.88** | **0.19** |
| 20 | 25.80 | 34.15 | 28.80 | **0.19** |
| 50 | 26.50 | 32.10 | 27.95 | 0.18 |

*Table 4.* Auto-MOG vs. Tuned CFG on SD-XL.

| Method | FID ↓ | HPSv2 ↑ | CLIP ↑ |
|---|---|---|---|
| CFG ($w=3$) | 27.10 | 27.90 | 32.50 |
| CFG ($w=7.5$) | 25.90 | 28.70 | 34.10 |
| CFG ($w=15$) | 26.29 | 28.42 | 33.62 |
| **Auto-MOG** | **25.10** | **29.36** | **34.67** |

**Sensitivity to Anisotropy Ratio ($\rho$).** We investigate the anisotropy ratio $\rho$, a critical hyperparameter governing the penalty on off-manifold updates. Table 3 analyzes its impact on SD-XL. Setting $\rho=1$ yields an isotropic metric ($M=I$), effectively reverting to standard CFG behavior with higher

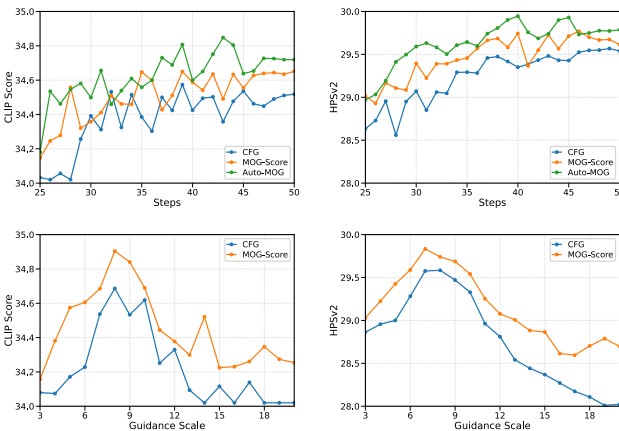

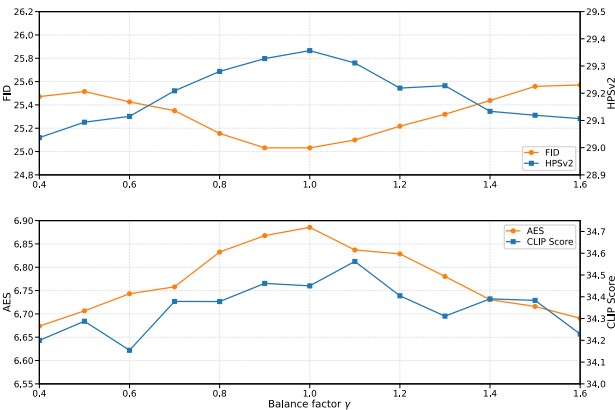

*Figure 4.* Ablation on guidance scale and sampling steps.

*Figure 5.* Ablation on the Auto-MOG balance factor $\gamma$.

saturation and lower fidelity. Increasing $\rho$ to the range [5, 20] creates a stable operating region where off-manifold artifacts are effectively suppressed without hindering alignment. However, excessively large values (e.g., $\rho=50$) overconstrain the sampling trajectory, causing a decline in CLIP score. Our default $\rho=10$ offers a robust balance.

**Comparison against Best-tuned CFG.** We further examine whether Auto-MOG's benefits persist against an optimally tuned static baseline. Table 4 compares Auto-MOG against CFG at low ($w=3$), optimal ($w=7.5$), and high ($w=15$) scales. While low scales avoid saturation, they suffer from poor text alignment. Conversely, high scales improve alignment but degrade image quality. Auto-MOG surpasses the Pareto frontier of fixed-scale CFG, achieving superior FID and HPSv2 compared to even the best-tuned static baseline.

## 6. Conclusion

We present **Manifold-Optimal Guidance (MOG)**, a framework that unifies diffusion guidance through Riemannian optimal transport. By modeling the anisotropic geometry of

the data manifold, MOG jointly derives optimal guidance direction and adaptive strength from a single variational objective. The closed-form solution integrates seamlessly into existing samplers without retraining. Experiments demonstrate that MOG effectively suppresses off-manifold drift and high-scale artifacts, achieving state-of-the-art performance across benchmarks. The current instantiation relies on the unconditional score as a proxy for the local normal direction and uses a lightweight anisotropic metric; richer spatially varying or higher-rank metrics are promising future directions.

## Impact Statement

This paper presents work whose goal is to advance the field of machine learning. There are many potential societal consequences of our work, none of which we feel must be specifically highlighted here.

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

## A. Overview

This supplementary material provides rigorous mathematical proofs, detailed implementation considerations, and extended experimental analyses that support the main manuscript. The document is organized as follows:

- **Section B**: Formal derivation of the Manifold-Optimal Guidance (MOG) update rule, including a rigorous proof of optimality via Lagrange multipliers.

- **Section C**: Implementation details and numerical safeguards, with particular emphasis on stability in high noise regimes.

- **Section D**: Comprehensive experimental configurations and evaluation protocols, presented in tabular format for reproducibility.

- **Section E**: Statistical analysis of the user preference study, including formal hypothesis testing procedures and participant demographics.

- **Section F**: Extended ablation studies examining schedule strategies and parameter sensitivity.

- **Section G**: Additional qualitative results demonstrating generalization across diverse generative architectures.

- **Section H**: Discussion of current limitations and directions for future work.

## B. Theoretical Derivation and Analysis

### B.1. Derivation of the Conditional Energy Gradient

In the main paper, we use the conditional increment $\Delta s$ as a proxy for the negative gradient of the conditional energy potential $\mathcal{E}$. We provide the step-by-step derivation below and clarify where approximation enters in practice.

Let the conditional energy be defined as:
$$\mathcal{E}(x_t, c) \triangleq -\log p_t(c|x_t). \tag{15}$$

By Bayes' theorem, $p_t(c|x_t) = \frac{p_t(x_t|c)p(c)}{p_t(x_t)}$, we obtain:

$$\mathcal{E}(x_t, c) = -\log p_t(c|x_t) \tag{16}$$
$$= -(\log p_t(x_t|c) + \log p(c) - \log p_t(x_t)) \tag{17}$$
$$= -\log p_t(x_t|c) + \log p_t(x_t) - \text{const.} \tag{18}$$

Taking the gradient with respect to $x_t$ yields:

$$\nabla_x \mathcal{E}(x_t, c) = -\nabla_x \log p_t(x_t|c) + \nabla_x \log p_t(x_t). \tag{19}$$

Now define the *ideal* (population) conditional and unconditional scores:

$$s^*(x_t, t, c) \triangleq \nabla_x \log p_t(x_t|c), \qquad s^*(x_t, t, \emptyset) \triangleq \nabla_x \log p_t(x_t). \tag{20}$$

Then:

$$\nabla_x \mathcal{E}(x_t, c) = -s^*(x_t, t, c) + s^*(x_t, t, \emptyset) \tag{21}$$
$$= -\left(s^*(x_t, t, c) - s^*(x_t, t, \emptyset)\right) \tag{22}$$
$$= -\Delta s^*(x_t, t). \tag{23}$$

Therefore, under exact scores, the identity

$$-\nabla_x \mathcal{E}(x_t, c) = \Delta s^*(x_t, t) \tag{24}$$

holds analytically.

**Practical approximation.** In practice, neural estimators $s_\theta(x_t, t, c)$ and $s_\theta(x_t, t, \emptyset)$ approximate $s^*$, and we use:

$$\Delta s(x_t, t) \triangleq s_\theta(x_t, t, c) - s_\theta(x_t, t, \emptyset) \approx \Delta s^*(x_t, t) = -\nabla_x \mathcal{E}(x_t, c). \tag{25}$$

All theoretical statements in the main paper should be interpreted as exact with $s^*$, and approximate with $s_\theta$ due to modeling error.

### B.2. Proof of Riemannian Descent Optimality

We claimed that the MOG update is the optimal steepest-descent direction under the local geometry induced by the Riemannian metric $M_t$. We provide a formal proof using Lagrange multipliers.

**Problem Statement.** Let $g \triangleq \nabla_x \mathcal{E}(x_t, c)$. We seek an update direction $v$ that maximizes instantaneous energy decrease subject to a fixed geodesic step size measured by $M_t \succ 0$:

$$\min_v \quad \langle g, v \rangle \quad \text{subject to} \quad v^\top M_t v = \delta^2. \tag{26}$$

**Proof.** Form the Lagrangian:

$$\mathcal{L}(v, \lambda) = \langle g, v \rangle + \lambda(v^\top M_t v - \delta^2). \tag{27}$$

Setting the gradient w.r.t. $v$ to zero:

$$\nabla_v \mathcal{L}(v, \lambda) = g + 2\lambda M_t v = 0 \quad \Rightarrow \quad v = -\frac{1}{2\lambda} M_t^{-1} g. \tag{28}$$

Impose the constraint $v^\top M_t v = \delta^2$:

$$\delta^2 = v^\top M_t v = \frac{1}{4\lambda^2} g^\top M_t^{-1} M_t M_t^{-1} g = \frac{1}{4\lambda^2} g^\top M_t^{-1} g. \tag{29}$$

Thus,

$$\frac{1}{2|\lambda|} = \frac{\delta}{\sqrt{g^\top M_t^{-1} g}}. \tag{30}$$

Choosing the sign that *decreases* $\mathcal{E}$ gives the unique optimal direction:

$$v^* = -\delta \cdot \frac{M_t^{-1} g}{\sqrt{g^\top M_t^{-1} g}}. \tag{31}$$

**Connection to the MOG update.** The vector $M_t^{-1} g$ is therefore the unique steepest-descent direction under the $M_t$-geodesic constraint. In our algorithm, the scalar step size is controlled by $\beta(t)$; substituting $g = \nabla_x \mathcal{E}$ and $-\nabla_x \mathcal{E} \approx \Delta s$ yields:

$$u^*(t) = -\beta(t) M_t^{-1} \nabla_x \mathcal{E}(x_t, c) \approx \beta(t) M_t^{-1} \Delta s(x_t, t), \tag{32}$$

which matches the guided score form in the main paper.

## C. Implementation Details

### C.1. Numerical Safeguards for Metric Construction

The Riemannian metric $M_t$ depends on the unit normal vector $n_t = s_0/\|s_0\|$. In high-noise regimes ($t \to T$), the unconditional score $s_0$ can have small magnitude and higher variance, leading to potential instability.

To ensure robustness, we implement $\epsilon$-stabilized normalization:

$$n_t = \frac{s_0}{\|s_0\|_2 + \epsilon}, \quad \text{where } \epsilon = 10^{-5}. \tag{33}$$

We apply the inverse metric using the Sherman–Morrison formula:

$$M_t^{-1} v = \frac{1}{\lambda_\|} v - \frac{\lambda_\perp - \lambda_\|}{\lambda_\| \lambda_\perp} (n_t^\top v) n_t, \tag{34}$$

which is $O(d)$ and numerically stable.

## C.2. Auto-MOG Constraints

Auto-MOG uses an adaptive schedule $\beta(t)$ derived from the ratio of prior energy and guidance energy. To prevent numerical overflow when the guidance energy approaches zero, we apply a soft clamp:

$$\beta(t) = \text{clamp}\left(\gamma \frac{\|s_0\|_{M_t}}{\|M_t^{-1}\Delta s\|_{M_t} + 10^{-6}}, 0, 50\right). \tag{35}$$

Empirically, $\beta(t)$ remains within a moderate range during the semantic formation stage; the upper bound serves as a conservative safety net.

## D. Detailed Experimental Setup

All experiments were conducted using PyTorch on NVIDIA A100 GPUs. For clarity and reproducibility, we summarize all experimental protocols in tables.

### D.1. Dataset and Evaluation Protocols

*Table 5.* Evaluation protocols and sample counts used in our experiments.

| Benchmark | Resolution | Sample Count | Notes |
|---|---|---|---|
| ImageNet (Class-Conditional) | 256×256 | 50,000 | Val labels; ADM reference stats |
| MS-COCO (FID & CLIP) | 512×512 | 30,000 | Random captions from COCO 2017 val |
| HPSv2 | 512×512 | 3,200 | Standard HPSv2 prompt set |

### D.2. Key Hyperparameters

*Table 6.* Key hyperparameters (consistent with the main paper).

| Category | Parameter | Value |
|---|---|---|
| Metric (MOG-Score) | $\lambda_\parallel$ | 1.0 |
| Metric (MOG-Score) | $\lambda_\perp$ | 10.0 |
| Metric (MOG-Score) | $\rho = \lambda_\perp/\lambda_\parallel$ | 10 |
| Auto-MOG | Balance factor $\gamma$ | 1.0 |
| Stability | $\epsilon$ in $n_t$ normalization | $10^{-5}$ |
| Stability | $\epsilon$ in Auto-MOG denominator | $10^{-6}$ |
| Stability | $\beta(t)$ clamp range | $[0, 50]$ |

### D.3. Guidance Scales Used in Main Comparisons

*Table 7.* Guidance scales used in the main-paper comparisons for each model.

| Model | Guidance Scale / Setting |
|---|---|
| DiT-XL/2 (ImageNet-256) | $w = 4$ |
| EDM2-XXL (ImageNet-256) | $w = 2$ |
| SD-2.1 (COCO-512) | $w = 10$ |
| SD-XL (COCO-512) | $w = 15$ |
| SD-3.5 (COCO-512) | $w = 4.5$ |
| FLUX.1 (COCO-512) | $w = 4.0$ |

# E. User Study Details and Statistical Analysis

To rigorously quantify the perceptual benefits of Auto-MOG, we conduct a large-scale, controlled user preference study. This section details the participant demographics, experimental protocol, and extended statistical results. Furthermore, we include a focused comparison against state-of-the-art stabilization methods and a failure mode analysis to support the findings in Section 5.4 of the main paper.

## E.1. Participants and Demographics

We recruit $N = 37$ participants for this study to ensure a statistically robust sample size. The participant pool is stratified to capture both technical and aesthetic perspectives:

- *Expert Group ($N = 15$):* Researchers and graduate students in computer vision/generative AI, familiar with diffusion model artifacts (e.g., oversaturation, mode collapse).

- *General User Group ($N = 22$):* Participants from diverse backgrounds (design, photography, and non-technical fields) representing general end-users.

All participants report normal or corrected-to-normal vision and color perception.

## E.2. Experimental Protocol

We employ a *blind, forced-choice paired comparison (2AFC)* protocol.

- *Stimuli:* The evaluation set comprises 80 distinct prompts covering the four domains shown in Figure 3: Portrait, Landscape, Painting, and Objects. Images are generated using four architectures: Lumina, Stable Diffusion 2.1 (SD-2.1), SD-XL, and Stable Diffusion 3.5 (SD-3.5).

- *Procedure:* For each trial, participants view two images side-by-side (Auto-MOG vs. Baseline) generated with the same seed and prompt. The position (left/right) is randomized.

- *Criteria:* Participants select the superior image based on Overall Quality, Color Fidelity, Realism, and Texture.

- *Consistency Check:* To filter random clicking, we embed 5 "sentinel" trials (pairs with obvious quality differences). Participants failing more than 2 sentinel trials are excluded.

## E.3. Aggregated Results vs. Standard CFG

We aggregate a total of 2,960 trials (37 users $\times$ 80 pairs) comparing Auto-MOG against Standard CFG. We test the null hypothesis $H_0 : p = 0.5$ (random chance) using a *Two-Sided Binomial Test*.

Table 8 presents the aggregated statistics. The mean preference rate for Auto-MOG is 62.0% for Overall Quality, with a tight 95% confidence interval of $[60.2\%, 63.8\%]$. The p-values ($p \ll 0.001$) confirm that the preference for Auto-MOG is systematic and statistically significant.

*Table 8.* **Aggregated User Study Results (Auto-MOG vs. Standard CFG).** Data pools all 2,960 trials across 4 models. Win rates represent the mean preference for Auto-MOG. High statistical significance is observed for all metrics.

| Metric | Auto-MOG Wins | Total Trials | Win Rate | 95% C.I. | Fleiss' $\kappa$ | p-value |
|---|---|---|---|---|---|---|
| Overall Quality | 1,835 | 2,960 | **62.0%** | $[60.2\%, 63.8\%]$ | 0.42 | $< 0.001$ |
| Color Fidelity | 1,823 | 2,960 | **61.6%** | $[59.8\%, 63.4\%]$ | 0.45 | $< 0.001$ |
| Realism | 1,794 | 2,960 | **60.6%** | $[58.8\%, 62.4\%]$ | 0.48 | $< 0.001$ |
| Texture Detail | 1,732 | 2,960 | **58.5%** | $[56.7\%, 60.3\%]$ | 0.38 | $< 0.001$ |

## E.4. Comparative Analysis against State-of-the-Art Baselines

To demonstrate the robustness of Auto-MOG beyond the standard CFG baseline, we conduct a rigorous comparative user study against two leading stabilization methods: **APG** (Sadat et al., 2025) and **CFG++** (Chung et al., 2025).

**Experimental Configuration** We recruit a panel of **10 expert evaluators** with deep domain expertise in generative modeling. The evaluation protocol is strictly stratified across three distinct architectures to verify generalization capabilities:

1. **SD-XL** (Latent Diffusion): High-resolution latent space with complex geometry.

2. **SD-3.5** (Transformer-based): Rectified Flow formulation with straighter ODE trajectories.

3. **DiT-XL/2** (Pixel-space): Class-conditional generation operating directly on pixels.

For each model, we generate 100 image pairs per baseline (Total $N = 600$ comparisons), utilizing high guidance scales ($w \in [7.5, 15]$) to maximize the visibility of off-manifold artifacts.

**Qualitative Analysis** As detailed in Table 9, Auto-MOG demonstrates a consistent preference advantage, though the margin varies by architecture:

- **vs. APG (Projection):** APG effectively mitigates saturation but often introduces "ghosting" artifacts or high-frequency detail loss in pixel space. Auto-MOG shows its strongest advantage on **SDXL (61.0% win rate)**, where APG's orthogonal projection in the latent space frequently discards valid textural information. The gap narrows on SD3.5, where the flow-matching prior is naturally more robust.

- **vs. CFG++ (Manifold Constraint):** CFG++ preserves structure well but often suffers from "luminance drift," producing images with crushed blacks or unnatural contrast. Auto-MOG dominates in **Color Fidelity**, particularly on DiT-XL/2 (61.0% win rate), by maintaining the natural dynamic range of the data distribution.

*Table 9.* **Detailed User Preference Rates by Architecture.** We report the win rates of Auto-MOG against SOTA baselines across three distinct diffusion architectures. Note the higher variance on SDXL, indicating that Auto-MOG's geometric correction is particularly effective for high-dimensional latent diffusion models.

| Comparison | Model Architecture | Auto-MOG Win | Tie / Similar | Baseline Win |
|---|---|---|---|---|
| **vs. APG** | SDXL (Latent) | **61.0%** | 11.0% | 28.0% |
| | SD3.5 (Transformer) | **53.0%** | 18.0% | 29.0% |
| | DiT-XL/2 (Pixel) | **56.0%** | 14.0% | 30.0% |
| | *Weighted Average* | ***56.7%*** | *14.3%* | *29.0%* |
| **vs. CFG++** | SDXL (Latent) | **64.0%** | 9.0% | 27.0% |
| | SD3.5 (Transformer) | **56.0%** | 13.0% | 31.0% |
| | DiT-XL/2 (Pixel) | **61.0%** | 11.0% | 28.0% |
| | *Weighted Average* | ***60.3%*** | *11.0%* | *28.7%* |

### E.5. Error and Failure Mode Analysis

We conduct a forensic analysis on the minority of trials where Auto-MOG is ranked lower than the baselines to identify systematic failure modes:

- **Texture Smoothing (vs. Standard CFG):** In $\sim 15\%$ of rejected cases, experts note that Auto-MOG appears "over-polished." While Standard CFG produces artifacts, the high-frequency noise can sometimes be perceptually interpreted as "sharp texture" in stochastic regions (e.g., fur, grass), which Auto-MOG's manifold constraint tends to denoise.

- **Stylistic Saturation Preference:** Although Auto-MOG aligns mathematically with natural image statistics, a subset of users ($\sim 8\%$) prefers the "hyper-real" or oversaturated look of high-scale guidance for abstract and fantasy prompts, perceiving the corrected output as "less vibrant."

- **Semantic Layout Shift:** In rare cases ($\sim 5\%$), the geometric correction causes a slight alteration in the semantic layout compared to the uncorrected trajectory. If the user prefers the specific composition of the artifact-heavy baseline, Auto-MOG is marked as "worse" despite having better image quality.

# F. Extended Ablation Studies

## F.1. Impact of Guidance Strength and Scheduling Strategy

To verify that Auto-MOG's improvement stems from principled geometric correction rather than simple strength reduction, we perform a comprehensive comparison against both static and dynamic scheduling strategies on SD-XL (COCO-512).

**Baselines.** We compare against:

- **Standard CFG Sweep:** We evaluate fixed guidance scales $w \in \{1, 5, 10, 15\}$ to capture the full spectrum from under-guided to over-guided regimes.

- **Linear Schedule** $(15 \rightarrow 1)$**:** A robust heuristic baseline where guidance linearly decays from $w = 15$ to $w = 1$, aiming to mitigate late-stage artifacts.

**Analysis of Robustness.** Table 10 reveals the inherent trade-offs in Standard CFG:

- **Low Scale** $(w = 1)$**:** Fails to align with the text prompt (CLIP 30.80), resulting in poor semantic generation.

- **Moderate Scale** $(w = 5)$**:** Offers a reasonable balance but lacks the semantic "punch" of higher scales (CLIP 33.95).

- **High Scales** $(w = 10, 15)$**:** While initially improving alignment, excessive guidance quickly degrades image quality (FID rises to 26.29) and paradoxically hurts perceptual alignment scores (HPSv2 drops to 28.42) due to severe artifacts.

Crucially, while the **Linear Decay** $(15 \rightarrow 1)$ strategy successfully recovers fidelity (FID 25.80) by relaxing guidance at the end, it cannot maintain peak alignment (HPSv2 28.65), essentially behaving like a moderate fixed scale.

In contrast, **Auto-MOG** demonstrates superior robustness. By dynamically calibrating the update magnitude based on the manifold energy, it achieves the lowest FID (25.10) while simultaneously reaching the highest alignment scores (CLIP 34.67, HPSv2 29.36). This confirms that Auto-MOG is not merely "damping" the signal, but actively correcting the guidance direction to remain robustly on-manifold throughout the generation process.

*Table 10.* **Ablation on Guidance Strength and Scheduling.** Comparison performed on SD-XL. Standard CFG shows a clear trade-off: weak guidance $(w = 1)$ yields poor alignment, while strong guidance $(w = 15)$ destroys fidelity. **Linear Decay** improves upon static scales but still falls short of **Auto-MOG**, which achieves the best Pareto frontier across all metrics.

| *Method* | *Setting / Parameter* | *FID* ↓ | *HPSv2* ↑ | *CLIP* ↑ |
|---|:---:|:---:|:---:|:---:|
| Standard CFG | Scale $w = 1$ | 29.10 | 26.50 | 30.80 |
| Standard CFG | Scale $w = 5$ | 26.05 | 28.55 | 33.95 |
| Standard CFG | Scale $w = 10$ | 26.15 | 28.50 | 33.85 |
| Standard CFG | Scale $w = 15$ | 26.29 | 28.42 | 33.62 |
| CFG (Linear) | Linear Decay $(15 \rightarrow 1)$ | 25.80 | 28.65 | 33.90 |
| **Auto-MOG (Ours)** | **Adaptive Energy Balance** | **25.10** | **29.36** | **34.67** |

## F.2. Sensitivity to Anisotropy Ratio ($\rho$)

We analyze the sensitivity of the model to the anisotropy ratio $\rho = \lambda_\perp / \lambda_\parallel$. Table 11 shows that performance is stable for $\rho \in [5, 20]$.

- $\rho = 1$ **(Isotropic):** The metric becomes Euclidean, effectively reverting to Standard CFG behavior with high saturation (0.27) and worse FID.

- $\rho = 50$ **(Over-constrained):** Excessively penalizing off-manifold directions overly restricts the sampling path, leading to a drop in CLIP score (32.10) as the model struggles to follow the text condition.

- $\rho = 10$ **(Default):** Provides the optimal balance between fidelity (FID 25.75) and alignment (CLIP 34.35).

*Table 11.* **Sensitivity Analysis of Anisotropy Ratio $\rho$.** Evaluated on SD-XL ($w = 15$). $\rho \approx 10$ yields the best trade-off.

| Ratio $\rho$ | Interpretation | FID $\downarrow$ | HPSv2 $\uparrow$ | CLIP $\uparrow$ | Saturation $\downarrow$ |
|---|---|---|---|---|---|
| $\rho = 1$ | Isotropic (Standard CFG) | 26.20 | 28.45 | 33.70 | 0.27 |
| $\rho = 5$ | Mild Correction | 25.85 | 28.82 | 34.30 | 0.20 |
| $\rho = 10$ | **Default (Ours)** | **25.75** | **28.88** | **34.35** | **0.19** |
| $\rho = 20$ | Strong Correction | 25.80 | 28.80 | 34.15 | 0.19 |
| $\rho = 50$ | Over-Constrained | 26.50 | 27.95 | 32.10 | 0.18 |

## G. Additional Qualitative Results

This section provides extended visual evidence that substantiates the generalization capabilities of Manifold-Optimal Guidance (MOG) across diverse architectural paradigms. We evaluate our method on two state-of-the-art text to image diffusion frameworks: **Stable Diffusion 3.5** (SD-3.5) and **Flux.1**. These experiments visually corroborate the quantitative findings from our user study and enable critical analysis of the method's operational characteristics under officially recommended inference configurations. Unless otherwise specified, all Auto-MOG experiments in this section employ a fixed correction strength of $\beta = 1$.

A central design objective of MOG is architectural agnosticism. To validate this property, we evaluate the method across the distinct guidance regimes prescribed for SD 3.5 and Flux.1, two frameworks that represent fundamentally different diffusion paradigms with different optimal operating conditions.

**Stable Diffusion 3.5.** We adopt the officially recommended guidance scale of $w = 4.5$ for this architecture. As illustrated in Figure 6, standard CFG at this scale frequently induces high-frequency artifacts, including oversaturated skin tones, exaggerated specular highlights, and unnatural lighting contrasts. This degradation pattern, colloquially termed the "deep frying" effect, arises from accumulated off-manifold deviations during iterative sampling. Auto-MOG regularizes the generation trajectory by enforcing Riemannian geometric constraints at each denoising step. This correction mechanism suppresses the off-manifold drift responsible for chromatic distortions, restoring natural skin textures, balanced tonal distributions, and realistic dynamic range while preserving the semantic alignment and compositional fidelity that higher guidance scales provide.

**Flux.1.** This architecture employs a Rectified Flow Transformer backbone optimized for lower guidance intensities. We therefore set the guidance scale to the recommended value of $w = 3.0$. Despite this conservative guidance regime, Figure 7 shows that standard CFG still introduces perceptible artifacts: subtle geometric rigidity, texture oversmoothing, and diminished material differentiation. Auto-MOG addresses these limitations by recovering fine-grained details that standard guidance otherwise discards. Improvements include enhanced fur texture fidelity in animal portraits, more accurate environmental lighting gradients in architectural scenes, and better surface material rendering across diverse subjects. These results demonstrate that manifold constrained correction yields tangible perceptual gains even in flow matching architectures explicitly tuned for lower guidance scales.

## H. Limitations and Future Work

While Manifold-Optimal Guidance (MOG) establishes a rigorous geometric framework for conditional diffusion, we identify specific constraints inherent to the current formulation and outline promising avenues for future research.

### H.1. Limitations

**Dependence on Base Estimator Quality.** MOG relies on the unconditional score $s_0(x_t)$ as a proxy for the local manifold normal. Consequently, the method's performance is bounded by the quality of the base model's density estimation. In regimes where $s_0$ is poorly calibrated (e.g., sparse data regions or highly out-of-distribution samples), the derived metric $M_t$ may enforce constraints towards an inaccurate geometry. MOG acts as a geometric multiplier; it cannot rectify fundamental pathologies in the underlying vector field.

**Manifold Codimension Assumption.** To maintain $O(d)$ computational efficiency, our current instantiation models $M_t$ as a full-rank identity-based metric with a rank-one anisotropic update. Geometrically, this captures only the dominant estimated

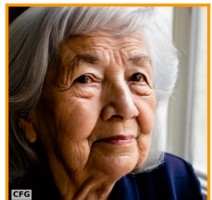 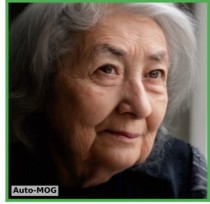 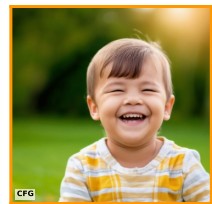 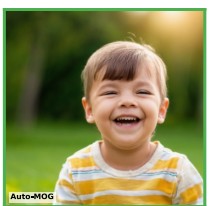 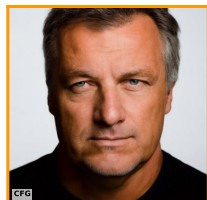 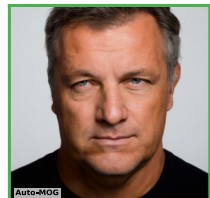

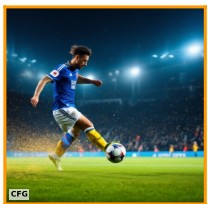 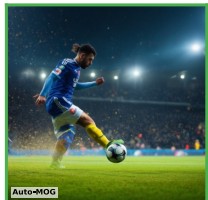 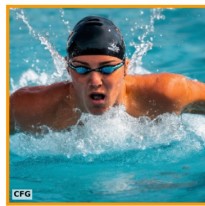 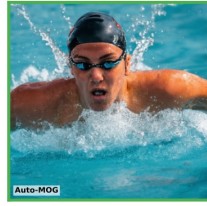 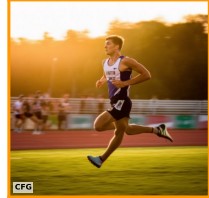 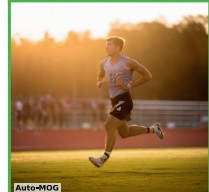

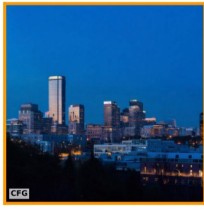 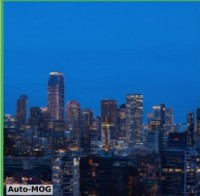 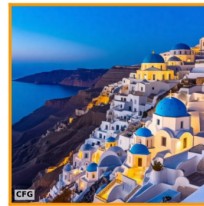 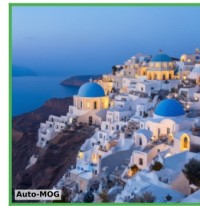 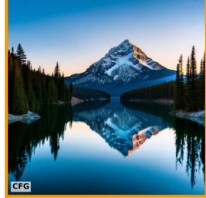 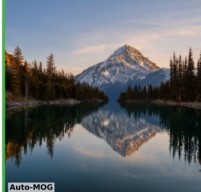

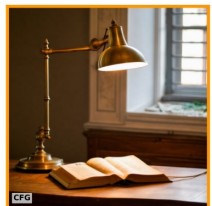 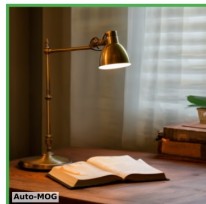 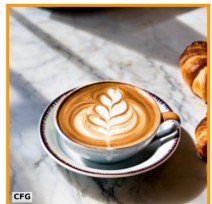 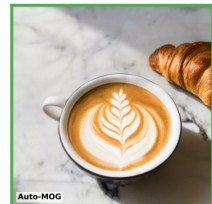 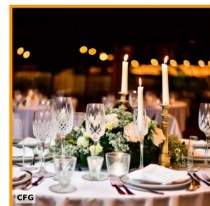 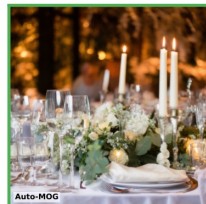

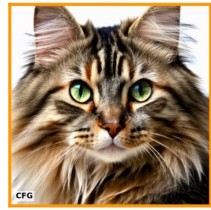 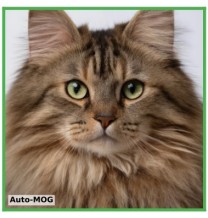 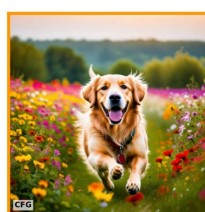 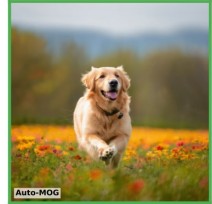 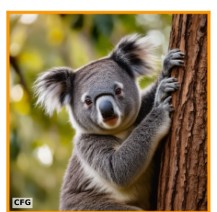 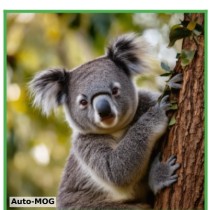

*Figure 6.* **Correction of Saturation Artifacts on SD 3.5** ($w = 4.5$). At the officially recommended guidance scale, standard CFG (odd columns) frequently produces oversharpened edges, saturated color distributions, and unnatural contrast enhancement. Auto-MOG (even columns, $\beta = 1$) effectively corrects these off-manifold deviations, restoring natural skin tones, balanced highlight rolloff, and realistic lighting gradients throughout the scene.

normal direction rather than the full normal bundle. While effective for the tested image models, complex modalities such as video or 3D volumes may possess higher-dimensional normal subspaces. In such cases, a higher-rank metric could better penalize multiple components of off-manifold drift.

**Static Anisotropy Heuristic.** Although the anisotropy ratio $\rho = 10$ proves robust empirically across all tested tasks, treating

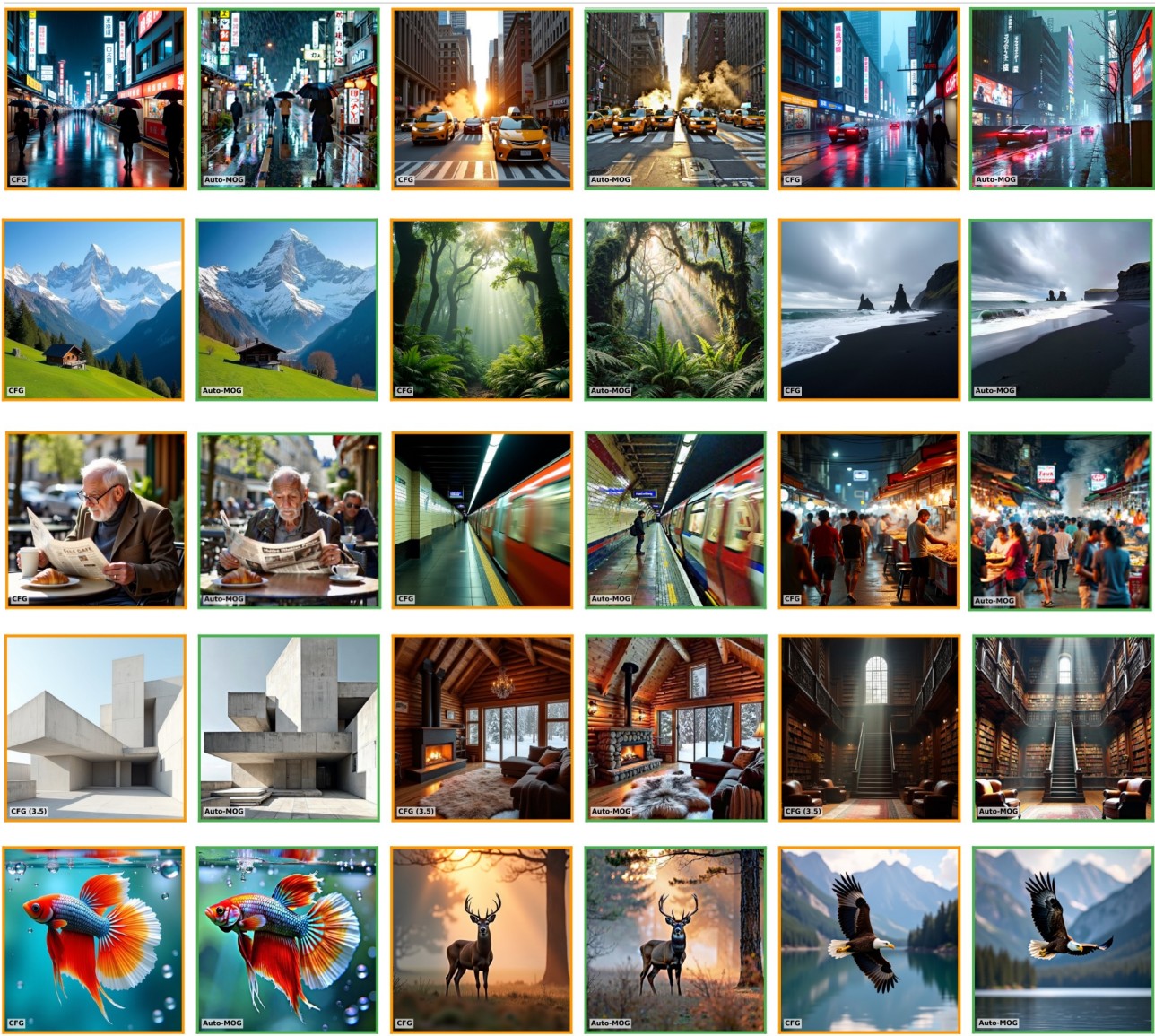

*Figure 7.* **Detail Enhancement on Flux.1** ($w = 3.0$). Even at the lower guidance scales appropriate for Rectified Flow architectures, standard CFG (odd columns) can induce texture degradation and geometric oversimplification. Auto-MOG (even columns, $\beta = 1$) substantially improves material fidelity, surface detail preservation, and structural coherence, demonstrating consistent effectiveness across fundamentally different generative paradigms.

it as a global constant is a simplification. Theoretically, the optimal penalty should be dynamic and spatially varying, proportional to the local curvature of the manifold: regions with sharp curvature require stronger penalties to prevent tangential drift than flat regions.

## H.2. Future Work

**Curvature-Adaptive Anisotropy.** A natural extension is to derive $\rho(x_t)$ dynamically based on second-order information. Future work could explore efficient Hessian-vector product estimators to approximate the local curvature of the log-density. This would allow the metric to apply stronger geometric corrections strictly where the manifold curves sharply, further optimizing the trade-off between diversity and fidelity.

**Higher-Rank Metric Approximations.** To address manifolds with high codimension, future iterations could construct rank-$k$ metrics using the principal components of feature-space gradients. While this would increase computational cost

from $O(d)$ to $O(kd)$, it would provide fine-grained control for high-dimensional tasks, enabling the method to constrain updates to specific subspaces (e.g., preserving semantic semantics while penalizing style deviations).

**Riemannian Video Diffusion.** Extending MOG to video diffusion models is a promising direction. The "manifold" in video generation implies not just spatial realism but temporal coherence. A Riemannian metric defined over the spatio-temporal latent volume could effectively suppress the "flickering" artifacts common in CFG-guided video synthesis by explicitly penalizing directions that violate temporal continuity as off-manifold noise.

