**Supplementary Material for**

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

 the metric $M_t$ as a rank-1 modification of the identity. Geometrically, this implicitly assumes that the "off-manifold" direction

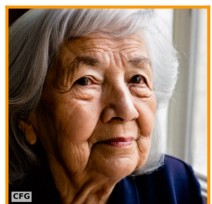 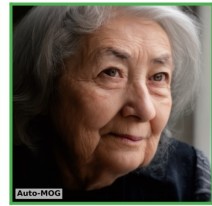 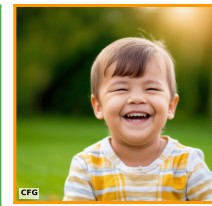 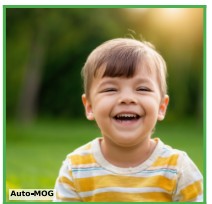 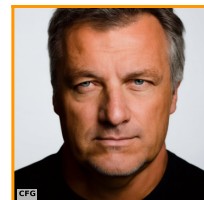 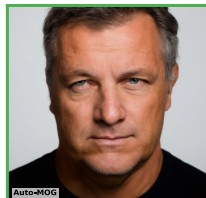

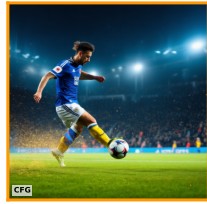 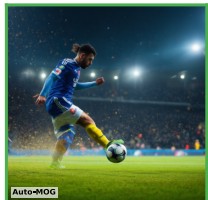 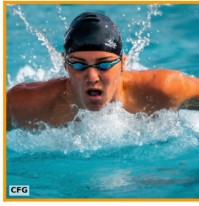 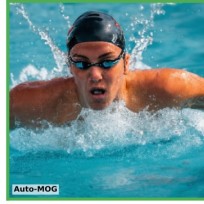 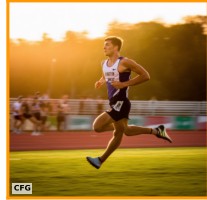 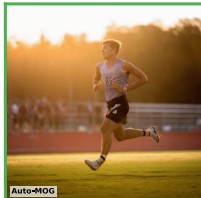

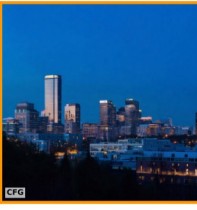 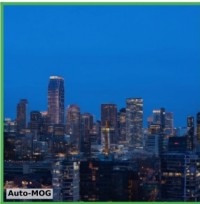 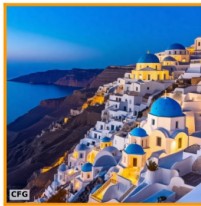 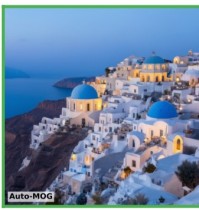 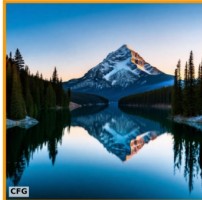 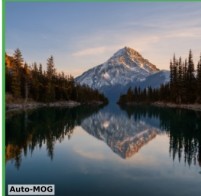

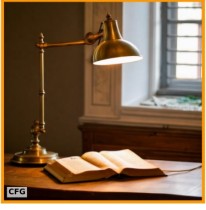 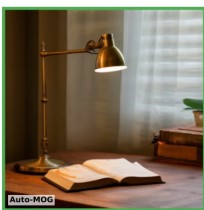 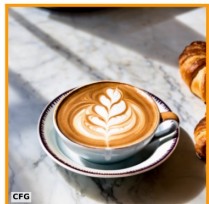 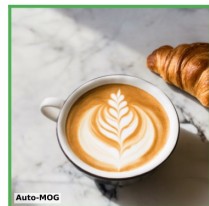 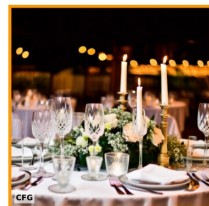 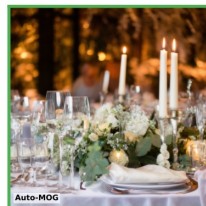

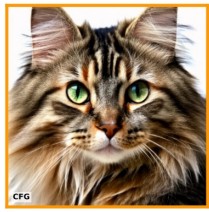 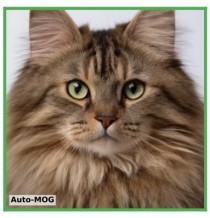 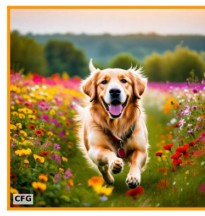 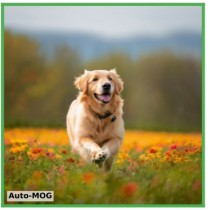 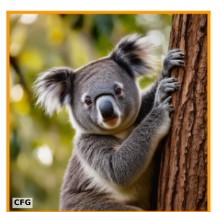 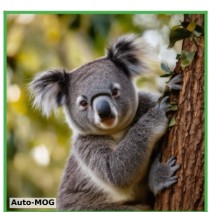

*Figure 6.* **Correction of Saturation Artifacts on SD 3.5** ($w = 4.5$). At the officially recommended guidance scale, standard CFG (odd columns) frequently produces oversharpened edges, saturated color distributions, and unnatural contrast enhancement. Auto MOG (even columns, $\beta = 1$) effectively corrects these off manifold deviations, restoring natural skin tones, balanced highlight rolloff, and realistic lighting gradients throughout the scene.

is locally one-dimensional (i.e., codimension 1). While sufficient for natural images, complex modalities (such as video or 3D volumes) may possess higher-dimensional normal subspaces. In such cases, a rank-1 metric might fail to penalize all components of off-manifold drift.

**Static Anisotropy Heuristic.** Although the anisotropy ratio $\rho = 10$ proves robust empirically across all tested tasks, treating

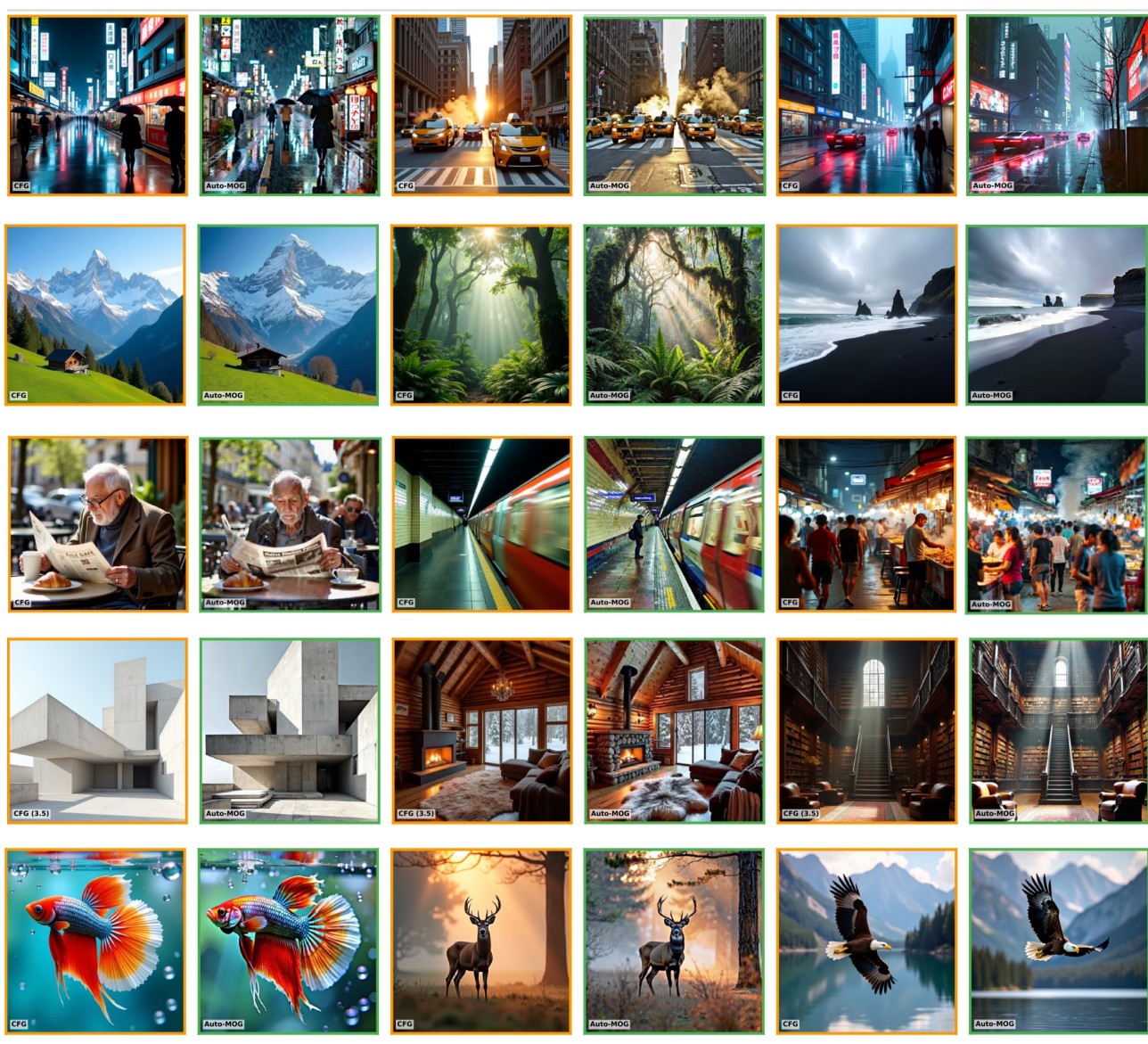

*Figure 7.* **Detail Enhancement on Flux.1** ($w = 3.0$). Even at the lower guidance scales appropriate for Rectified Flow architectures, standard CFG (odd columns) can induce texture degradation and geometric oversimplification. Auto MOG (even columns, $\beta = 1$) substantially improves material fidelity, surface detail preservation, and structural coherence, demonstrating consistent effectiveness across fundamentally different generative paradigms.

it as a global constant is a simplification. Theoretically, the optimal penalty should be dynamic and spatially varying, proportional to the local curvature of the manifold: regions with sharp curvature require stronger penalties to prevent tangential drift than flat regions.

## H.2. Future Work

**Curvature-Adaptive Anisotropy.** A natural extension is to derive $\rho(x_t)$ dynamically based on second-order information. Future work could explore efficient Hessian-vector product estimators to approximate the local curvature of the log-density. This would allow the metric to apply stronger geometric corrections strictly where the manifold curves sharply, further optimizing the trade-off between diversity and fidelity.

**Higher-Rank Metric Approximations.** To address manifolds with high codimension, future iterations could construct rank-$k$ metrics using the principal components of feature-space gradients. While this would increase computational cost

from $O(d)$ to $O(kd)$, it would provide fine-grained control for high-dimensional tasks, enabling the method to constrain updates to specific subspaces (e.g., preserving semantic semantics while penalizing style deviations).

**Riemannian Video Diffusion.** Extending MOG to video diffusion models is a promising direction. The "manifold" in video generation implies not just spatial realism but temporal coherence. A Riemannian metric defined over the spatio-temporal latent volume could effectively suppress the "flickering" artifacts common in CFG-guided video synthesis by explicitly penalizing directions that violate temporal continuity as off-manifold noise.