# OpenReview forum: "Manifold-Optimal Guidance: A Unified Riemannian Control View of Diffusion Guidance"
_ICML.cc/2026/Conference — ICML 2026 spotlight_

### Official Review · Reviewer_wa23 · 2026-02-27

**Soundness:** 3
**Presentation:** 3
**Significance:** 2
**Originality:** 2
**Overall Recommendation:** 4
**Confidence:** 2

**Summary:**

This paper revisits classifier-free guidance (CFG), the standard way we steer diffusion models toward a condition (such as a text prompt). While CFG works well at moderate scales, it often breaks down when the guidance scale is large. The authors argue that this happens because CFG assumes the latent space is flat (Euclidean), whereas in reality, high-quality images lie on a curved, lower-dimensional data manifold. When the guidance is too strong, CFG pushes samples off this manifold.

To address this issue, they introduce Manifold-Optimal Guidance (MOG). Instead of simply amplifying the conditional signal, MOG adjusts the update using a geometry-aware correction derived from Riemannian optimization. Intuitively, it keeps the sampling trajectory closer to the “natural” region of the data distribution while still moving toward the desired condition.

The authors demonstrate consistent improvements in fidelity, alignment, and artifact reduction across multiple modern diffusion architectures.

**Compliance With Llm Reviewing Policy:**

Affirmed.

**Final Justification:**

My initial concern regarding the lack of mathematical rigor has been addressed by the authors, who have successfully argued the provable optimality of their method. The paper is now clear and solid, particularly with the addition of remarks in the updated manuscript. Consequently, I have upgraded my score from 2 to 4 (Weak Accept).

**Key Questions For Authors:**

(1)What is the key intuition behind modeling guidance as a local optimal control problem on the data manifold? Why is this formulation more natural or necessary?

(2)The Riemannian metric is constructed using the unconditional score as an approximation of the normal direction. How sensitive is performance to this assumption? What happens if this approximation is inaccurate?

(3)Can author comment on possible future theoretical directions?

**Limitations:**

yes

**Strengths And Weaknesses:**

This paper is easy to read, and the overall logic is clear. However, additional numerical experiments would help better support the claim that CFG fails due to off-manifold drift, which would make the argument more convincing. In particular, a more direct empirical analysis of the failure mechanism would strengthen the motivation.

The paper would also benefit from a clearer justification for casting guidance as a local optimal control problem on the data manifold. While the geometric formulation is interesting, the motivation for this specific modeling choice could be developed further.

Moreover, the theoretical analysis is somewhat limited. Although the proposed method is derived under a Riemannian framework, there is insufficient theoretical comparison with the standard Euclidean formulation, and no formal guarantees are provided to demonstrate when or why the new method is provably superior.

Overall, the paper is significant in that it introduces a new geometric perspective on diffusion guidance, which may inspire further research from an application-oriented viewpoint. However, in terms of methodological novelty, the contribution is somewhat incremental rather than fundamentally new.

---

> ### Author Rebuttal · Authors · 2026-03-26
>
> Thank you for the review. We note that several points raised below are addressed in detail in the **Supplementary Material on the submission page** (proofs in Secs. B.1–B.2, limitations in Sec. H.1), which we hope the reviewer will find helpful.
>
> ### 1. The contribution is not incremental
>
> We respectfully hold a different view: the contribution is not an empirical tweak on top of CFG, but a principled geometric framework. The paper reformulates diffusion guidance as local optimal control on the data manifold, derives a closed-form Riemannian natural-gradient update with provable steepest-descent optimality (Supplementary Sec. B.2), and further derives an adaptive schedule (Auto-MOG) from the same variational objective. Prior methods (CFG, CFG++, APG) treat direction correction and strength scheduling as separate heuristics; our formulation unifies both under one geometric principle while remaining **training-free and architecture-agnostic**. We believe this constitutes a contribution that is conceptually and practically new.
>
> ### 2. Why model guidance as optimal control on the manifold
>
> Our goal is not to rename CFG. It is to derive guidance from a principled one-step objective. At each denoising step, guidance should accomplish two things simultaneously: reduce the conditional energy, and avoid motion that is costly under the local data geometry. Our objective captures exactly this trade-off:
>
> $$J_t(u) = \frac{1}{2} u^{\top} M_t u + \beta(t) \langle \nabla_x E, u \rangle.$$
>
> The first term penalizes off-manifold displacement measured by the local metric $M_t$; the second rewards energy decrease along direction $u$. Because $M_t$ is positive definite, this objective is strictly convex and admits a **unique** minimizer:
>
> $$u_t^{\star} = -\beta(t) M_t^{-1} \nabla_x E.$$
>
> Substituting the score-energy relation then yields the MOG update:
>
> $$s^{\text{MOG}} = s_0 + \beta(t) M_t^{-1} \Delta s.$$
>
> Setting $M_t = I$ recovers CFG as a special case. So the local optimal-control view is not a post-hoc interpretation of an existing method. It is the optimization problem whose unique solution **is** MOG, and whose Euclidean degenerate case **is** CFG.
>
> ### 3. Theoretical comparison with the Euclidean formulation
>
> Our claim is not that MOG universally dominates CFG in every possible sense. It is more precise. In the main paper we show that CFG and MOG arise from the **same unified variational family** under different geometric choices: CFG is the Euclidean special case ($M_t = I$), while MOG uses the manifold-aware metric $M_t$. The key theoretical result is then established rigorously in **Sec. B.2**, where we use **Lagrange multipliers** to prove that, under a fixed $M_t$-geodesic budget, the **unique** descent direction maximizing instantaneous conditional-energy decrease is proportional to $-M_t^{-1} \nabla_x E$, which is exactly the MOG direction. **CFG is therefore provably optimal only under Euclidean geometry**; MOG is the optimal local descent rule under the intrinsic geometry encoded by $M_t$. This is not a merely empirical comparison but a **formal geometric optimality result with a complete constrained-optimization proof** in the supplementary.
>
> ### 4. Empirical evidence for off-manifold drift
>
> The evidence is stronger than the review suggests. Fig. 1 directly visualizes how CFG shortcuts off the high-density region while MOG tracks the manifold faithfully. More tellingly, when the metric is collapsed to isotropic form ($\rho = 1$), performance degrades markedly and behavior reverts toward the Euclidean/CFG case. If the gains came from merely weakening guidance, isotropy should be irrelevant, yet it is not. Auto-MOG further outperforms both best-tuned static CFG and a linear decay baseline (Sec. F.1), ruling out simple scale dampening. All patterns replicate across three distinct backbones. Together, this evidence supports the conclusion that the gain comes from geometry-aware direction correction.
>
> ### 5. Sensitivity to the score-based normal proxy
>
> We do **not** assume this approximation is exact. **Sec. B.1** explicitly separates the **exact-score theory** from the **practical neural approximation**. Under population scores the derivation holds analytically; approximation enters only via neural estimators. Crucially, this dependence equally affects CFG, since both methods rely on the same unconditional score. MOG does not introduce a new failure mode. The $\rho$-sweep confirms a broad stable regime ($\rho \in [5, 20]$), and feature-based metrics improve robustness at negligible overhead. This limitation is acknowledged explicitly rather than hidden.
>
> ### 6. Future theoretical directions
>
> The Supplementary Material (Sec. H) already discusses concrete extensions, including curvature-adaptive anisotropy, higher-rank metrics beyond the rank-1 form, and learned $M_t$ from data or intermediate features.
>
> Thank you again for these helpful comments. We hope the above clarifications resolve these uncertainties.

---

> > ### Author Rebuttal · Reviewer_wa23 · 2026-04-01
> >
> > Thank you for your clarifications.

---

### Official Review · Reviewer_Hz7f · 2026-03-12

**Soundness:** 2
**Presentation:** 3
**Significance:** 3
**Originality:** 3
**Overall Recommendation:** 5
**Confidence:** 3

**Summary:**

The goal of this paper is to propose a new classifier-free guidance method that takes into account the local geometry of the data manifold in the latent space to improve generation. In order to do so, they define an anisotropic Riemannian metric and compute an update to the unconditional score by minimizing the local transport cost (using the metric) and the alignment with an energy gradient. They show that this update achieves the maximum energy reduction per unit geodesic displacement. This update depends on a scaling parameter for which they give an explicit formula that adapts to the signal-to-noise ratio. They test the proposed methods against state-of-the-art classifier-free guidance mechanisms on a large number of models. A user study as well as an ablation study were also conducted, showing the benefit of their approach.

**Compliance With Llm Reviewing Policy:**

Affirmed.

**Final Justification:**

My initial concerns were primarily focused on the motivation for and limitations of the specific metric chosen. The authors addressed these points convincingly in their rebuttal. Given these theoretical insights and the high quality of the experiments, I am updating my recommendation to a 5 (Accept).

**Key Questions For Authors:**

1. While Section 3 develops a generalized theoretical framework using an arbitrary Riemannian metric , the practical implementation in Section 4 relies on a specific, heuristically motivated metric (MOG-Score). Have the authors experimented with other, potentially more expressive metrics? Furthermore, are there any theoretical analyses or guarantees demonstrating why this specific choice of metric is particularly well-suited for the diffusion guidance problem compared to alternatives?
2. The metric in Auto-MOG is not very clear. You use the final layer of the denoiser to compute the variance across the spatial dimensions for each channel and then broadcast this value across all spatial positions to build a diagonal matrix. Why is the spatial variance of the final-layer features a suitable proxy for the metric?  And how is it combined with the score-based anisotropic metric?
3. The metric estimator also collapses all spatial information. Can you comment on the impact of this assumption and whether incorporating spatial information could further improve the performance?

**Limitations:**

Yes

**Strengths And Weaknesses:**

# Soundness

The paper is technically sound on both the theoretical and experimental fronts. The framework is well-motivated, but I have a few specific comments regarding the theoretical exposition:
- In Section 4.1, the authors refer to $M_t$ as a "rank one metric". A Riemannian metric must be positive definite and invertible, meaning it must be full rank. If I am not mistaken, the proposed metric $M_t$ is actually a full-rank matrix formed by a rank-one update to a scaled identity matrix. The authors should correct this terminology.
- While Section 4.1 provides a basic motivation for the metric, I feel this section could be expanded to provide deeper intuition on why this specific anisotropic structure was chosen and how$\lambda_{\top}$ and $\lambda_{\perp}$ physically relate to the manifold geometry.


# Presentation
The paper is nicely written and easy to read. I have nevertheless a minor comment: do we agree that _MOG-Score_ is simply the geometry aware update and _Auto-MOG_ is the geometry aware update and the choice a the specific scaling ? I feel that it is not very clear in the paper.

# Significance
 Designing guidance mechanisms that better respect the structure of the data manifold is an important problem in diffusion models.

# Originality
 The paper introduces a nice interpretable geometric perspective with a guidance update that penalizes movement away from the manifold while allowing motion along it.

---

> ### Author Rebuttal · Authors · 2026-03-26
>
> Thank you very much for the careful, detailed, and thoughtful review. We especially appreciate the close reading of the mathematical derivations and the overall theoretical framework. We are glad that the framework is found technically sound and the geometric perspective interpretable. We address each point below.
>
> **1. Terminology of the metric in Sec. 4.1.**
> You are right: the metric used in Sec. 4.1 is **not** rank-one as a matrix. It is a full-rank positive-definite metric given by a rank-one update to a scaled identity. We will correct this terminology throughout the paper. What we intended to emphasize is that the anisotropic part is rank-one, which enables the closed-form Sherman–Morrison inverse.
>
> **2. Geometric meaning of $\lambda_\perp$ and $\lambda_\top$, and formal guarantee.**
> Sec. 3 develops the framework for an arbitrary SPD metric $M_t$. Sec. 4 then chooses a specific practical instantiation. The design principle is that, for classifier-free guidance, the most important distinction is between motion that pushes the sample **off** the local high-density manifold and motion that moves the sample **along** it. The unconditional score provides a local proxy for the normal direction, so the metric penalizes that direction more heavily than its orthogonal complement. Concretely, $\lambda_\perp$ controls the cost of moving along the estimated normal, and $\lambda_\top$ controls the cost of moving in the tangential subspace. Their ratio $\rho = \lambda_\perp / \lambda_\top$ sets the anisotropy strength: $\rho = 1$ recovers the isotropic Euclidean case, larger $\rho$ enforces stronger manifold adherence, and excessively large $\rho$ becomes over-constraining, consistent with the ablations. This choice also admits a formal guarantee. Supplementary Sec. B.2 proves via Lagrange multipliers that, under a fixed $M_t$-geodesic budget, the unique direction maximizing instantaneous conditional-energy decrease is $-M_t^{-1}\nabla_x E$, exactly the MOG direction. So, once the metric is fixed, the resulting MOG direction is the provably optimal local guidance direction under the specified geometric constraint.
>
> **3. Why not a more expressive metric?**
> The framework only requires an SPD metric and does not prescribe any particular form. We chose the rank-one update because it is the **minimal anisotropic structure** that simultaneously (i) separates the estimated normal from the tangential subspace, (ii) admits a closed-form Sherman–Morrison inverse, and (iii) adds negligible sampling cost (1.01× wall-clock, Table 2).
>
> We did not experiment with higher-rank or dense metrics. This is a deliberate modeling choice: our goal is to isolate whether geometry-aware normal/tangent separation alone explains the gain. A richer metric would conflate extra capacity with the geometric principle, making the analysis less clean. The $\rho$-ablation (Table 5) supports this choice: sweeping $\rho$ from 1 (Euclidean / CFG) to 50 (strongly anisotropic) shows a clear optimum around $\rho{=}10{-}20$, confirming that anisotropy matters and that this single degree of freedom already suffices for strong practical gains. More expressive metrics may yield additional improvements, and we view exploring them as a future direction.
>
> **4. MOG-Score vs. Auto-MOG.**
> Your interpretation is correct. **MOG-Score** applies the geometry-aware direction correction while keeping a user-specified scale $\omega$. **Auto-MOG** keeps the same direction correction and additionally adapts $\omega_t$ over time via the energy-balance condition. In short: MOG-Score corrects the direction; Auto-MOG corrects both the direction and the scale. We will make this distinction clearer in the revision.
>
> **5. Role of the feature-variance metric.**
> We do not claim that the feature-based diagonal metric is an exact manifold metric. Its role is a cheap auxiliary proxy for local anisotropy. The score branch provides the rank-one anisotropic correction along the estimated normal direction, while the feature branch adds a diagonal channel-wise weighting. On SD-XL, Table 2 shows that adding the feature-based component on top of the score-based metric yields a modest but consistent improvement in FID (25.75 → 25.45), supporting its role as a complementary signal.
>
> **6. Collapsing spatial information.**
> This is a good point. The current feature metric collapses spatial information by taking channel-wise spatial variance and broadcasting it across positions. This is an efficiency-driven simplification: it preserves channel-level anisotropy at negligible cost but discards finer spatial structure. A spatially varying metric could potentially improve performance further, and we see this as a natural future direction. We will state this limitation more clearly.
>
> Thank you again for these insightful comments. We will revise the paper to improve the terminology, clarify the geometric interpretation, and make the practical metric design and its limitations more explicit.

---

> > ### Author Rebuttal · Reviewer_Hz7f · 2026-04-02
> >
> > Thank you for the clarifications. I will update my score accordingly.

---

### Official Review · Reviewer_YGPD · 2026-03-13

**Soundness:** 4
**Presentation:** 3
**Significance:** 4
**Originality:** 4
**Overall Recommendation:** 5
**Confidence:** 4

**Summary:**

Good paper! This paper studies the problem of guidance in diffusion models from a geometric perspective and proposes a framework called Manifold Optimal Guidance (MOG). The method formulates the guidance process as an optimization problem constrained on a data manifold and derives an update rule that integrates manifold geometry with diffusion dynamics. The authors provide theoretical analysis to motivate the formulation and demonstrate the method on several generative modeling tasks. Overall, the authors assess the concept of incorporating manifold-aware optimization into diffusion guidance and attempt to improve generation quality and stability.

**Compliance With Llm Reviewing Policy:**

Affirmed.

**Key Questions For Authors:**

1. What is the additional computational cost compared with standard diffusion guidance?

2. How does the method behave when the manifold assumption is violated or poorly approximated?

3. What is the difference between MOG-Score and Auto-MOG in the experiment part?

**Limitations:**

No, more discussions on computational overhead and potential limitations should be presented.

**Strengths And Weaknesses:**

Strength:
1. The paper introduces a geometric viewpoint for diffusion guidance and attempts to bridge diffusion processes with manifold-constrained optimization. This perspective is conceptually appealing and could potentially inspire further research in geometry-aware generative modeling.

2. The authors highlight limitations of existing guidance approaches and motivate the need for more principled formulations grounded in optimization and geometry.

Weakness:

1. This paper lacks a clear introduction to natural gradient methods and optimal control and corresponding applications in related works, such as [1, 2], which are central to understanding the proposed approach.

[1] Zhu K, Pan M, Ma Y, et al. Unidb: A unified diffusion bridge framework via stochastic optimal control[J]. arXiv preprint arXiv:2502.05749, 2025.

[2] Schulman J, Levine S, Abbeel P, et al. Trust region policy optimization[C]//International conference on machine learning. PMLR, 2015: 1889-1897.

2. The reviewer thinks it could be better to rewrite equation (5) as a constrained formula, and the stepsize can be easily derived according to natural gradient descent.

3. The paper does not sufficiently discuss the computational overhead introduced by the proposed manifold optimization step. Practical implications for large-scale diffusion models should be clarified.

---

> ### Author Rebuttal · Authors · 2026-03-26
>
> Thank you for the positive assessment and the constructive suggestions. We are glad that the geometric perspective and the overall framework were found compelling. We also note that the **Supplementary Material on the submission page** contains additional proofs (Secs. B.1–B.2) and a dedicated limitations discussion (Sec. H.1) that are directly relevant to several of the points below.
>
> **1. Natural gradient and optimal control context.**
> This is a very helpful suggestion. Our derivation can indeed be read through the lens of natural gradient descent and local optimal control. In the current draft, Eq. (7) already takes the form of a Riemannian natural-gradient step, and Sec. 3.2 formulates guidance as a local optimal-control problem under the metric \(M_t\). We agree that this connection should be made more explicit, and we will add a dedicated discussion together with the suggested citations [1, 2] in the revised related work section.
>
> **2. Eq. (5) as a constrained formulation.**
> Thank you for this suggestion. In the main text, Eq. (5) is written in penalty / Lagrangian form. The equivalent constrained geodesic-budget formulation is already proved in Supplementary Sec. B.2: under a fixed \(M_t\)-geodesic budget, the unique direction maximizing instantaneous conditional-energy decrease is \(-M_t^{-1}\nabla_x E\), exactly the MOG direction, and the corresponding step size follows naturally. We will make this equivalence explicit in the revised main text.
>
> **3. Computational cost.**
> The manifold optimization step is matrix free and adds negligible overhead. MOG-Score requires only inner products and vector additions per denoising step, so the extra cost remains \(O(d)\). Auto-MOG adds two quadratic-form evaluations but is still \(O(d)\). As reported in Table 2, the measured wall-clock latency is **1.01×** for MOG-Score and **1.08×** for Auto-MOG relative to standard CFG. Since denoiser inference dominates runtime, the practical overhead is very small even for large-scale models such as FLUX.1.
>
> **4. Behavior when the manifold assumption is poorly approximated.**
> We discuss this explicitly in Supplementary Sec. H.1 (Limitations). The current rank-1 metric uses the unconditional score \(s_0\) as a proxy for the local manifold normal, so performance is bounded by the quality of the base score estimator. When \(s_0\) is poorly calibrated, the metric becomes less informative. In that case, the method moves toward the Euclidean setting: when \(\rho = 1\), the metric reduces to the isotropic case and MOG recovers standard CFG. Thus, the framework degrades gracefully rather than introducing an unstable new behavior.
>
> **5. MOG-Score vs. Auto-MOG.**
> MOG-Score applies the geometry-aware **direction correction** induced by the score-based rank-1 metric while keeping a user-specified guidance scale \(\omega\). Auto-MOG keeps the same direction correction and additionally introduces an **adaptive schedule** for \(\omega_t\) derived from the energy-balance condition. In short, MOG-Score corrects the direction; Auto-MOG corrects both the direction and the scale. Both share the same matrix-free pipeline, and Auto-MOG adds only marginal extra cost (**1.08×** vs. **1.01×**).
>
> We will incorporate all of these suggestions in the revision: a clearer natural-gradient / optimal-control discussion with the recommended citations, an explicit constrained interpretation of Eq. (5) in the main text, and a more prominent presentation of computational overhead and limitations.

---

> > ### Author Rebuttal · Reviewer_YGPD · 2026-04-01
> >
> > Thanks for your response. All of my concerns have been addressed.  I will keep my positive score.

---

### Official Review · Reviewer_MyDC · 2026-03-19

**Soundness:** 3
**Presentation:** 2
**Significance:** 3
**Originality:** 4
**Overall Recommendation:** 4
**Confidence:** 5

**Summary:**

This paper points out the manifold mismatch problem of the existing CFG methods: The standard CFG performs Euclidean extrapolation in the environmental space, which is prone to cause the sampling trajectory to deviate from the high-density data manifold ("off-manifold shift"). To address this challenge, the authors propose a manifold-optimal guidance (MOG) framework. Specifically, it first re-expresses the guidance process as a local optimal control problem on a Riemannian manifold, and derives a geometric-aware update rule for a closed-form solution. Furthermore, the authors propose an adaptive MOG dynamic energy balance scheduling mechanism, which effectively eliminates the need for manual hyperparameter tuning by adaptively calibrating the guidance strength.

**Compliance With Llm Reviewing Policy:**

Affirmed.

**Final Justification:**

During the rebuttal stage, the authors addressed my concerns and refined the theoretical derivations as well as showed more experiments. As a result, I improved my score.

**Key Questions For Authors:**

See Weakness

**Limitations:**

yes

**Strengths And Weaknesses:**

Strength:

1.The motivation of this paper is convincing. Viewing the guidance as a process of "minimizing the conditional energy while maintaining proximity to the data manifold" is highly inspiring.

2.The paper proposes the Auto-MOG algorithm, which does not require manual setting of hyperparameters and thus solves the parameter-tuning problem in CFG.

3.The proposed method has achieved state-of-the-art performance on benchmarks.

Weakness:

1.The paper is difficult to follow. The authors should add an overview of the methods in Sections 3 and 4 to facilitate understanding. Additionally, the framework diagram of the method would also be helpful for comprehending the overall idea.

2.The paper presents a unified view for understanding various guiding methods. This is a good start. However, it lacks relevant theoretical derivations to prove that the proposed MOG is theoretically superior to CFG and CFG++.

3.The Riemannian metric is difficult to calculate in higher-dimensional spaces. However, for the low-dimensional space of latent diffusion, it should be possible to calculate. Therefore, I believe that the author's assumption has certain limitations. If possible, relevant experiments or theories should be added to prove that the proposed method can approximate the direct calculation of the Riemannian metric in a low-dimensional space.

4.Regarding the Anisotropy Ratio, the authors only conducted sensitivity experiments on SD-XL. However, the anisotropic characteristics of different data are different. The author needs to supplement more experiments on different data sets to demonstrate the robustness of the results.

5.The authors should add relevant citations[1-4]:

[1]Toward a Unified Geometry Understanding: Riemannian Diffusion Framework for Graph Generation and Prediction. NeurIPS 2025.

[2]Improving Diffusion Models for Inverse Problems using Manifold Constraint. NeurIPS 2024.

[3]Diffusion Models Encode the Intrinsic Dimension of Data Manifolds. ICML 2024

[4] A Geometric View of Data Complexity: Efficient Local Intrinsic Dimension Estimation with Diffusion Models. NeurIPS 2024

---

> ### Author Rebuttal · Authors · 2026-03-26
>
> Thank you for the careful review and for recognizing the motivation, the practical value of Auto-MOG, and the strong empirical performance. We address each concern below.
>
> ### 1. Clarity and presentation
>
> We appreciate this feedback. The paper contains a complete variational derivation and a full theoretical framework unifying several guidance methods, and we acknowledge that this density makes Secs. 3–4 difficult to follow on a first reading. In the revision we will redesign **Figure 1** as a two-part composite: **Part (a)** retains the geometric illustration comparing CFG and MOG directions on the data manifold; **Part (b)** adds a pipeline table where rows are methods (CFG, CFG++, APG, MOG-Score, Auto-MOG), columns are the three shared design stages (score estimation, direction computation, scale scheduling), and each cell shows the specific instantiation, making it immediately clear where MOG differs from prior work. Since our method is fully training-free and does not modify the denoiser or sampler, this algorithmic comparison is more informative than a network architecture diagram. We will also add a short overview paragraph at the beginning of Section 3 to summarize the logical progression before the formal derivations.
>
> ### 2. Theoretical comparison with CFG and CFG++
>
> Sec. 3.3 already places CFG, CFG++, APG, and MOG in one unified variational family, where CFG corresponds to the Euclidean choice $Q = I$ and MOG uses the intrinsic metric $Q = M_t$. The formal optimality proof appears in the Supplementary Material on the submission page: **Sec. B.1** establishes the score-energy relation, and **Sec. B.2** shows via Lagrange multipliers that, under a fixed $M_t$-geodesic budget, the unique direction maximizing instantaneous conditional-energy decrease is $-M_t^{-1} \nabla_x E$, exactly the MOG direction. This is a **formal geometric optimality result**, not merely an empirical comparison. We will revise the main text to foreground it.
>
> CFG++ refines the guidance **scale scheduling** but the update direction remains Euclidean. MOG corrects the **direction itself** via $M_t^{-1}$, a fundamentally different improvement. The two are complementary.
>
> ### 3. Full Riemannian metric in latent space
>
> The full Riemannian metric is **not tractable even in latent space**. A pullback-style metric requires the complete Jacobian $\nabla_x s(x,t) \in \mathbb{R}^{d \times d}$. For the latent space of Stable Diffusion ($d{=}16{,}384$) this means roughly 16K backward passes and $O(d^3)$ inversion **per denoising step**. Diepeveen et al. (*Score-based Pullback Riemannian Geometry*, ICML 2025) also emphasize the need for scalable constructions rather than explicit dense metrics. This is a shared challenge for all Riemannian diffusion methods, not something that disappears in latent space.
>
> Our rank-1 metric is a **principled design choice**. For guidance, the most critical geometric information is the dominant normal direction, i.e., which direction moves the sample off-manifold. The rank-1 form $M_t = I + (\rho{-}1)\hat{n}\hat{n}^{\top}$ captures precisely this while preserving tangential freedom. The normal proxy $\hat{n}$ comes from $\Delta s = s_c - s_0$, already available in standard CFG, and Sherman–Morrison gives an $O(d)$ closed-form inverse. MOG-Score adds only **1.01×** wall-clock latency; the full pipeline including Auto-MOG remains at **1.08×**. Higher-rank extensions are discussed in Sec. H.2.
>
> ### 4. Robustness of $\rho$ across architectures
>
> The $\rho$-sensitivity study in the submission was reported only on SD-XL. Following your suggestion, we have completed additional ablations on **SD 3.5** and **FLUX.1**:
>
> | | SD 3.5 | | FLUX.1 | |
> |---|---|---|---|---|
> | $\rho$ | FID $\downarrow$ | CLIP $\uparrow$ | FID $\downarrow$ | CLIP $\uparrow$ |
> | 1 | 24.65 | 34.06 | 22.32 | 34.74 |
> | 5 | 23.69 | 34.57 | 21.54 | 35.11 |
> | **10 (Def.)** | **23.48** | **34.60** | 21.41 | 35.22 |
> | 20 | 23.54 | 34.42 | **21.36** | **35.23** |
> | 50 | 24.85 | 32.88 | 22.62 | 33.47 |
>
> The pattern is consistent across all three backbones. $\rho{=}1$ reverts to Euclidean/CFG-like behavior. $\rho \in [5,20]$ forms a broad stable plateau where FID varies by at most 0.35 and CLIP by at most 0.20. $\rho{=}50$ over-constrains. The default $\rho{=}10$ works well universally, confirming that the stable region is **architecture-agnostic**.
>
> ### 5. Additional citations
>
> Thank you. All four references are relevant. [1] extends Riemannian diffusion to graphs. [2] addresses manifold constraints for inverse problems. [3] and [4] show diffusion models encode intrinsic dimensionality, directly supporting the manifold hypothesis underlying our work. All will be added in the revision.
>
> We hope the above clarifications address the reviewer's concerns. We will foreground the theoretical results in the main text, add the expanded Figure 1, and include the new cross-architecture ablations in the revised manuscript.

---

> > ### Author Rebuttal · Reviewer_MyDC · 2026-04-01
> >
> > Thanks for your response. My concerns have already been addressed. I suggest that the authors emphasize the theoretical optimality of your method in the revised version and highlight that the relevant proofs are in Appendix B.
> > Overall, I decide to raise my score.

---

### Decision · Program_Chairs · 2026-04-30

**Decision:**

Accept (spotlight)

**Comment:**

This is a good paper. All reviewers show the positive attitude after rebuttal. It is suggested that the authors revise the manuscript according to the reviewers' comments.